# Why Deep Jacobian Spectra Separate: Depth-Induced Scaling and Singular-Vector Alignment

**Nathanaël Haas** [1 2]   **François Gatine** [3]   **Augustin Cosse** [2]   **Zied Bouraoui** [1]

## Abstract

Understanding why gradient-based training in deep networks exhibits strong implicit bias remains challenging, in part because tractable singular-value dynamics are typically available only for balanced deep linear models. We propose an alternative route based on two theoretically grounded and empirically testable signatures of deep Jacobians: depth-induced exponential scaling of ordered singular values and strong spectral separation. Adopting a fixed-gates view of piecewise-linear networks, where Jacobians reduce to products of masked linear maps within a single activation region, we prove the existence of Lyapunov exponents governing the top singular values at initialization, give closed-form expressions in a tractable masked model, and quantify finite-depth corrections. We further show that sufficiently strong separation forces singular-vector alignment in matrix products, yielding an approximately shared singular basis for intermediate Jacobians. Together, these results motivate an approximation regime in which singular-value dynamics become effectively decoupled, mirroring classical balanced deep-linear analyses without requiring balancing. Experiments in fixed-gates settings validate the predicted scaling, alignment, and resulting dynamics, supporting a mechanistic account of emergent low-rank Jacobian structure as a driver of implicit bias.

## 1. Introduction

Deep networks often develop an effectively low-rank Jacobian geometry: a small number of singular directions

[1]CRIL UMR 8188, Université d'Artois, CNRS, France [2]LMPA, Université du Littoral Côte d'Opale [3]IMJ-PRG, Sorbonne Université . Correspondence to: Nathanaël Haas <haas@cril.fr>, François Gatine <gatine@imj-prg.fr>.

*Proceedings of the 43rd International Conference on Machine Learning*, Seoul, South Korea. PMLR 306, 2026. Copyright 2026 by the author(s).

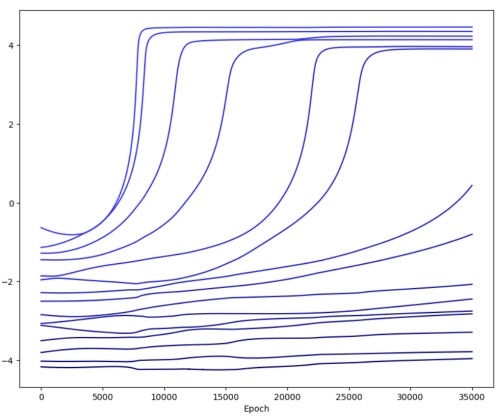

*Figure 1.* Evolution of the top-15 Jacobian log-singular values during training of a fixed-gates linear network (depth 10, width 64) on a synthetic rank-10 regression task. Inputs are Gaussian and targets are generated by a fixed random linear map of rank 10. Colors index order: brighter curves correspond to larger singular values (from $s_1$ to $s_{15}$).

dominate input–output sensitivity while the remainder stay weak. This behavior is typically accompanied by spectral separation, in the sense that ratios between consecutive ordered singular values increase, and it can sharpen during training; such anisotropy has been associated with generalization in realistic architectures (Oymak et al., 2019). Figure 1 and 2 illustrate this phenomenon on a rank-10 synthetic task for fixed-gates linear network and ReLU MLP without bias, where leading Jacobian singular values separate and evolve in a structured manner. These trajectories raise a mechanistic question: which depth-driven properties of Jacobian products create and maintain strong separation, and when do they enable tractable predictions for singular-value evolution?

A classical setting where singular-value evolution is analytically tractable is deep linear networks and matrix factorization. Exact learning dynamics and depth-dependent amplification effects are well understood in the linear case (Saxe et al., 2014), and gradient descent in factorized models exhibits implicit low-complexity bias (Gunasekar et al., 2017). Our starting point is the singular-value dynamics derived by Arora et al. (2019), which characterizes the evolution of ordered singular values of the end-to-end linear map under

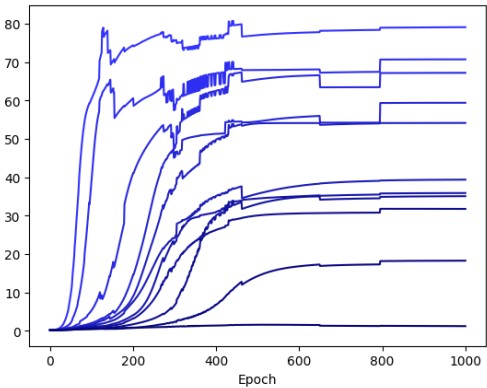

*Figure 2.* Evolution of the top-11 Jacobian log-singular values during training of a ReLU MLP (depth 3, width 64) on the same synthetic task as in Figure 1 task. The Jacobian is computed from the same sample across training. Colors index order: brighter curves correspond to larger singular values (from $s_1$ to $s_{15}$). We used a shallower network than in deep linear to prevent noise to visually overcome the overall dynamics. We see jumps on top of the dynamics of Figure 1 due to gate pattern changes during the training as expected.

gradient descent in the balanced regime. This result gives a clean spectral-level view of learning dynamics and serves as a baseline for mode-wise trajectories such as those in Figure 1. The balancing hypothesis is restrictive: it prevents direct consideration of standard random initialization and is tightly coupled to end-to-end linearity. It is therefore unclear how to use it as a mechanism for Jacobian spectra in more general architectures. In parallel, Jacobian spectra have been extensively studied through signal propagation and dynamical isometry at random initialization (Schoenholz et al., 2017; Pennington et al., 2017; 2018), and through empirical sensitivity measures during training (Novak et al., 2018). These lines of work illuminate trainability regimes and bulk or extremal spectral behavior, but they do not directly yield predictions for the evolution of ordered singular values at finite depth in a way that explains strong separation, nor do they provide a structural replacement for balancing that would enable deep-linear-like spectral dynamics.

Our goal is to recover deep-linear-like singular-value dynamics without assuming balancing, under conditions compatible with random initialization and with the gated-product structure of Jacobians. We argue that a simple depth-driven mechanism can substitute for balancing: (i) depth-induced scaling creates separation among ordered singular values, (ii) sufficiently strong separation promotes alignment of dominant singular vectors across products, and (iii) once leading directions stabilize, singular-value evolution becomes approximately decoupled. This yields deep-linear-like dynamics without balancing. We make this mechanism concrete using gated matrix products motivated by Jacobians of piecewise-linear networks. Conditioning on a fixed acti-

vation pattern, the Jacobian reduces exactly to a product of weight matrices interleaved with diagonal gates; we isolate this structure via fixed-gates and masked linear networks. Concretely, we (i) establish a depth-scaling characterization for masked products at random initialization, including explicit Lyapunov exponents in a tractable masked model and finite-depth corrections, yielding a principled source of spectral separation; (ii) show that strong spectral separation is sufficient to force alignment of dominant singular vectors in matrix products, implying an approximately shared singular basis across intermediate Jacobians; and (iii) combine these ingredients to motivate an approximation regime where singular-value evolution becomes effectively decoupled, mirroring the functional form of balanced deep-linear dynamics up to a global scaling. Our analysis justifies treating depth scaling and spectral separation as plausible structural conditions, and they guide the approximations that yield a deep-linear-like singular-value dynamic for fixed-gates networks without assuming balancing.

## 2. Related Works

**Jacobian spectra, signal propagation, and training-time geometry.** The spectral properties of the input–output Jacobian have been extensively studied, especially at random initialization. Signal propagation analyses characterize regimes of stability and chaos through Jacobian norms and singular value distributions in the infinite-width limit, leading to the notions of edge of chaos and dynamical isometry (Schoenholz et al., 2017; Pennington et al., 2017; 2018). These works primarily describe bulk statistics or extremal behavior via mean-field approximations, and they do not track ordered singular values individually or their finite-depth separation. Complementary empirical studies relate Jacobian sensitivity to generalization during training (Novak et al., 2018), and (Oymak et al., 2019) reports that after training, MLPs and CNNs often exhibit a spectrum with a few large singular values, linking this spectral separation to generalization. In contrast, we study ordered singular values at finite depth in gated product models and formalize sufficient conditions for spectral separation that do not rely on infinite-width limits or isometry assumptions.

**Products of random matrices and Lyapunov exponents.** The asymptotic behavior of singular values of products of random matrices is classically described by the multiplicative ergodic theorem (Furstenberg & Kesten, 1960; Oseledets, 1968), which guarantees the existence of Lyapunov exponents under suitable conditions, typically including invertibility. Precise characterizations of Lyapunov spectra for Gaussian products are known in several settings (Newman, 1986; Crisanti et al., 1993). However, these results generally do not apply directly to products involving rank-deficient or structured factors. Neural Jacobians often take

the form of products of matrices interleaved with diagonal gating operators, and gating can make these factors singular, placing them outside standard invertible-product theory. We extend Lyapunov-type reasoning to this gated setting by introducing conditioned $(r, p)$-gates, allowing to characterize depth-induced exponential scaling and the resulting spectral separation for masked linear networks at realistic depths.

**Implicit bias in deep linear and factorized models.** For deep linear networks, the implicit bias of gradient descent is well understood. Exact singular-value dynamics were derived by Saxe et al. (2014), revealing depth-dependent amplification and low-rank bias. Subsequent work generalized these insights to matrix and tensor factorizations, showing that gradient descent implicitly favors low-complexity solutions that can be related to nuclear norms or analogous measures (Gunasekar et al., 2017; Arora et al., 2019). These analyses rely crucially on linearity and on assumptions such as balancing that enforce alignment across layers. We recover a deep-linear-like mode-wise dynamics in the presence of fixed gating, but without assuming balanced initialization. Related recent evidence suggests that low-complexity effects induced by factorization may also matter in modern architectures, for example through connections between nuclear-norm-like biases and out-of-context generalization behavior in transformer settings (Huang et al., 2025).

**Gated and nonlinear architectures.** Recent works study implicit bias in models with gating or structured nonlinearities. Lippl et al. (2022) analyzes generalized gated linear networks and characterizes asymptotic solutions selected by gradient descent under shared-parameter constraints. Jacot (2023) introduces a function-space notion of rank and shows that infinitely deep nonlinear networks exhibit a bias toward low-rank functions. While these works provide asymptotic or equilibrium characterizations, they do not describe finite-depth, ordered spectral dynamics of Jacobians induced by fixed activation patterns. Our focus is instead on finite-depth behavior and on a mechanism in which spectral separation alone induces alignment of singular vectors in products, independently of balancing or infinite-depth limits.

**Alignment and spectral gaps.** Alignment phenomena in deep linear networks have been studied by Ji & Telgarsky (2019; 2020), who show that gradient descent aligns singular directions across layers under suitable conditions. Classical matrix perturbation theory provides tools for subspace stability under spectral gaps (Davis & Kahan, 1970; Stewart & Sun, 1990). We complement these perspectives by showing that strong spectral separation itself forces singular-vector alignment in matrix products, yielding quantitative convergence rates and providing the structural ingredient that underlies our approximate decoupled singular-value dynamics.

**Low-rank bias beyond weights.** Several works study rank minimization of network weights or explicit regularization of Jacobians (Timor et al., 2023; Galanti et al., 2025; Scarvelis & Solomon, 2024). These approaches promote low rank through architectural or algorithmic choices. Our contribution is orthogonal: low-rank structure and spectral separation can emerge implicitly in the input–output Jacobian due to depth-induced effects in gated products, even without explicit regularization.

## 3. Overview and Notations

We write $\log$ for the natural logarithm, so that $\log(e) = 1$. Unless specified otherwise, all matrices are square $n \times n$ real matrices. The subscript $M_{i,j}$ denotes the $(i, j)$-th entry of a matrix $M$. If $(M_i)$ is a sequence of matrices and $a \leq b$, let $M_{a:b} := M_b \ldots M_a$. If $a > b$, we set $M_{a:b} := I$ the identity matrix. In any singular value decomposition (SVD) $M = USV^\top$, the singular values $s_1(M), \ldots, s_n(M)$ (i.e. the coefficients of $S$) are ordered from highest to lowest. Our contribution is built upon the following blocks.

**Setup and baseline.** Section 4 introduces fixed-gates and masked linear networks as models for Jacobian products, and recalls the balanced deep-linear baseline that motivates our target dynamics (Proposition 4.5).

**Depth scaling at initialization.** Section 5 studies masked products at random initialization and proves depth scaling, including finite-depth corrections (Theorem 5.3). This block is self-contained and does not assume balancing.

**Alignment from spectral separation.** Section 6 isolates a matrix-product phenomenon: under spectral separation, dominant singular directions align across products (Theorem 6.1). This result is asymptotic in depth and is used as a structural ingredient in the synthesis.

**Approximate singular-value dynamics.** Section 7 combines the two structural ingredients, depth scaling and spectral separation-induced alignment, as assumptions to obtain a deep-linear-like singular-value evolution for fixed-gates networks up to a multiplicative factor (Proposition 7.1; see also Proposition 7.2). Theorems 5.3 and 6.1 are not chained into a fully rigorous training-time statement; rather, they motivate the approximation regime used in this section.

**Experiments.** Section 8 collects plots of experiments illustrating the validity of our results and of the subsequent assumptions they motivate.

Appendices follow the same order: proofs of the baseline dynamics (Appendix A), background on exterior powers (Appendix B), proofs of the main theorems and propositions (Appendices C–E), definition of diagonal correlation used in section 8 (Appendix F) and additional experiments (Appendix G).

## 4. Fixed-Gates Products as Jacobian Models

We introduce fixed-gates linear networks as controlled models for Jacobian products along fixed activation patterns in piecewise-linear networks. We then recall a balanced singular-value dynamics result for masked networks that motivates our target approximation.

**Definition 4.1** (Fixed-Gates Linear Network). Let $L \geq 1$. Let $W_\ell \in \mathbb{R}^{n_\ell \times n_{\ell-1}}$ for $\ell = 1, \ldots, L$ be trainable weight matrices, and let $D_\ell \in \mathbb{R}^{n_\ell \times n_\ell}$ for $\ell = 1, \ldots, L-1$ be fixed diagonal matrices (the *gates*). A *Fixed-Gates Linear Network (FGLN)* of depth $L$ is the linear map

$$J := W_L D_{L-1} W_{L-1} \cdots D_1 W_1.$$

For convenience, we set $D_0 := I_{n_0}$ and $D_L := I_{n_L}$ when needed. In the special case where each $D_\ell$ has diagonal entries in $\{0, 1\}$, we call it a *Masked Linear Network*. In the special case where each $W_\ell$ is square of size $n$, we call it a *square FGLN of width $n$*.

*Remark* 4.2. We allow rectangular layers. When we refer to singular values of partial products, they are taken with the usual SVD convention for rectangular matrices.

**Definition 4.3** (Multi-mode FGLN). Let $\mathcal{M}$ a set of modes. For each mode $m \in \mathcal{M}$, let

$$J^{(m)} := W_L D_{L-1}^{(m)} W_{L-1} \cdots D_1^{(m)} W_1$$

be an FGLN that shares the same weights $(W_\ell)_{\ell=1}^L$ but may have mode-dependent fixed gates $(D_\ell^{(m)})_{\ell=1}^{L-1}$. Let $\sigma$ be a mode function mapping each input $x$ to an index $\sigma(x) \in \mathcal{M}$. A *Multi-mode FGLN* is the pair $(\sigma, (J^{(m)})_{m \in \mathcal{M}})$, and it represents the piecewise-linear map $x \mapsto J^{(\sigma(x))} x$.

*Remark* 4.4. An MLP without biases is a Multi-Modes FGLN with a trainable mode matching function $\sigma$.

In the case of a Masked Linear Network (MLN) we can extend the analysis done on deep linear networks dynamics from (Arora et al., 2019). The proof, found in Appendix A, relies on the idempotency of gates of MLNs.

**Proposition 4.5** (Balanced singular-value dynamics for MLNs). *Consider a Masked Linear Network (MLN) $J(t) = W_L(t) D_{L-1} \cdots D_1 W_1(t)$ trained by gradient flow on the weights $(W_\ell(t))_{\ell=1}^L$ with loss $\mathcal{L}(J)$. Let $D_0 := I$ and $D_L := I$, and define $M_\ell(t) := D_\ell W_\ell(t) D_{\ell-1}$ for $\ell = 1, \ldots, L$. Assume the balancing hypothesis at initialization:*

$$M_{\ell+1}^\top(0) M_{\ell+1}(0) = M_\ell(0) M_\ell^\top(0), \qquad \ell = 1, \ldots, L-1.$$

*Under gradient flow, this balancing condition is preserved along training. Let $J(t) = \sum_k s_k(t) u_k(t) v_k(t)^\top$ be an SVD of $J(t)$. Then the singular values satisfy, for each $k$,*

$$\dot{s}_k(t) = -L \, s_k(t)^{2-2/L} \left\langle \nabla_J \mathcal{L}(J(t)), \, u_k(t) v_k(t)^\top \right\rangle,$$

*where $\langle A, B \rangle = \mathrm{Tr}(A^\top B)$ is the Frobenius inner product.*

Proposition 4.5 relies crucially on linearity and on a balancing invariant that is preserved by gradient flow. In the rest of the paper we take a different route by isolating two structural properties of deep gated products that can be stated and tested directly at the level of Jacobians.

**Definition 4.6** (Depth scaling). Let $M_{1:\ell} := M_\ell \cdots M_1$ denote partial products and let $\varepsilon \geq 0$. We say that a family of products satisfies $\varepsilon$-*depth scaling* at rank $r$ if there exist constants $(\gamma_k, \delta_k)_{k=1}^r$ such that for all $\ell \in \{1, \ldots, L\}$ and all $k \leq r$,

$$|\log s_k(M_{1:\ell}) - (\ell \gamma_k + \delta_k)| \leq \varepsilon.$$

**Definition 4.7** (Spectral separation: working definition). Let $\varepsilon > 0$. We say that a rectangular matrix $M$ exhibits $\varepsilon$-*spectral separation* at rank $r$ if for all $k \leq r$

$$\frac{s_{k+1}(M)}{s_k(M)} < \varepsilon$$

with the convention $s_{n+1}(M) = 0$.

The next section studies how depth scaling arises at random initialization in FGLNs and thus MLP Jacobians, how spectral separation induces alignment of singular directions in products, and how these ingredients lead to an approximate deep-linear-like singular-value evolution for FGLNs.

## 5. Spectral Separation and Depth Scaling at Initialization

In this section, $n \geq 1$ denotes a fixed integer. We consider $n \times n$ real matrices unless specified otherwise. Consider the gated product $J_L = W_L D_{L-1} W_{L-1} \cdots D_1 W_1$ at initialization. Denote $s_{i,L}$ its $i$-th singular value. Experimentally, $\frac{1}{L} \log s_{i,L}$ converges slowly to some value $\gamma_i$, called the *Lyapunov exponent*. When all gates $D_\ell$ are set as the identity matrix the existence of the exponents $\gamma_i$ is a known fact from random matrix theory, see for instance (Bougerol et al., 2012). More generally if $J_L$ is a product of random invertible i.i.d. matrices satisfying mild assumptions, one is able to show the existence of Lyapunov exponents. In general, the product $D_\ell W_\ell$ may not be invertible, preventing us from using this fact as a black box.

We prove the existence of Lyapunov exponents $\gamma_i$ for $J_L$ at initialization, and we compute them explicitly in a Bernoulli-gated Gaussian model. To account for finite-depth effects, we also derive a first-order correction in expectation:

$$\mathbb{E}\left[\frac{1}{L} \log s_{i,L}\right] = \gamma_i + \frac{d_{i-1} - d_i}{L} + o\left(\frac{1}{L}\right),$$

where the constants $d_i$ are universal in the sense that they depend only on $i$ (through a Haar-orthogonal minor) and not on $(p, \sigma)$. This correction yields a more accurate approximation for the top of the spectrum at moderate depth.

## 5.1. Terminology and Statement

**Definition 5.1.** Let $0 < p \leq 1$, and $\sigma > 0$.

- We say a random $n \times n$ matrix $W$ is $\sigma$-*Ginibre* if it has i.i.d. entries distributed as $\mathcal{N}(0, \sigma^2)$.

- We say a random $n \times n$ matrix $D$ is a $p$-*gate* if it is diagonal with i.i.d. Bernoulli($p$) diagonal entries.

Consider the product $J_L = M_L \cdots M_1$ where each layer is $M_\ell = D_\ell W_\ell$, with $D_\ell$ a $p$-gate and $W_\ell$ a $\sigma$-Ginibre matrix. When analyzing $J_L$ as $L \to \infty$ a technical degeneracy arises: almost surely, some gate becomes the zero matrix at a sufficiently large depth, forcing $J_L$ to be eventually zero. This phenomenon is irrelevant at the moderate depths and widths used in practice, where gates have rank concentrated near $np$. To obtain a mathematically meaningful infinite-depth limit while staying faithful to the practical regime, we condition gates to have rank at least $r$, leading to the $(r, p)$-gate model below.

**Definition 5.2.** Let $1 \leq r \leq n$ be an integer, $0 < p \leq 1$, and $\sigma > 0$.

- We say a random $n \times n$ matrix $D$ is an $(r, p)$-*gate* if it is distributed as a $p$-gate conditioned to having rank at least $r$.

- We say a random $n \times n$ matrix $M$ is an $(r, p, \sigma)$-*layer* if it is of the form $DW$, with $D$ an $(r, p)$-gate, $W$ a $\sigma$-Ginibre matrix, $D$ and $W$ independent.

Each diagonal entry of an $(r, p)$-gate $D$ follows a Bernoulli($p$) distribution, but these entries are *not* independent. However in practice $D$ almost behaves as a $p$-gate. Indeed, to sample an $(r, p)$-gate one simply has to sample successive $p$-gates and keep the first one with rank $\geq r$. For $r \ll np$, this holds immediately with very high probability.

We state our main theorem.

**Theorem 5.3.** *Let $1 \leq r \leq n$ be an integer, $0 < p \leq 1$, and $\sigma > 0$. Consider a sequence of i.i.d. $(r, p, \sigma)$-layers $(M_i)$, and write $M_i = D_i W_i$. For any positive integer $L$, define the random matrix*

$$J_L = M_L \cdots M_1 = (D_L W_L) \cdots (D_1 W_1)$$

*Denote $s_{1,L} \geq \cdots \geq s_{n,L}$ the singular values of $J_L$. Then there exists explicit real numbers called* Lyapunov expo-nents

$$\gamma_1 > \cdots > \gamma_r$$

*which depend on $r, p$ and $\sigma$ such that almost surely*

$$\frac{1}{L} \log(s_{i,L}) \xrightarrow{L \to \infty} \begin{cases} \gamma_i & \text{if } 1 \leq i \leq r \\ -\infty & \text{if } i > r. \end{cases}$$

*Moreover, for $1 \leq i \leq r$, the expectation satisfies*

$$\mathbb{E}\left[\frac{1}{L}\log(s_{i,L})\right] = \gamma_i + \frac{d_{i-1} - d_i}{L} + o\left(\frac{1}{L}\right)$$

*with*

$$d_i = -\mathbb{E}[\log|\det \Omega^{i,i}|]$$

*where $\Omega^{i,i}$ is the $i \times i$ upper left block of a Haar-distributed orthogonal matrix (and $d_0 = 0$).*

A proof of Theorem 5.3 is provided in Appendix C. Our argument is general; although the Theorem is stated for MLNs, the proof applies to many other choices of initialization for FGLN (Theorem C.1).

*Remark* 5.4 (Closed form for $(r, p, \sigma)$-layers with Ginibre weights). Under the assumptions of Theorem 5.3, for each $1 \leq i \leq r$ the Lyapunov exponent is

$$\gamma_i = \log(\sqrt{2}\sigma) + \frac{\sum_{t=r}^{n} \left[\binom{n}{t}p^t(1-p)^{n-t}\psi\left(\frac{t-i+1}{2}\right)\right]}{2\sum_{m=r}^{n}\binom{n}{m}p^m(1-p)^{n-m}},$$

where $\psi$ is the digamma function. For the non-masked case $p = 1$, we recover $\gamma_i = \log(\sqrt{2}\sigma) + \frac{1}{2}\psi(\frac{n-i+1}{2})$ as in (Newman, 1986)[1].

*Remark* 5.5 (High-probability regime $r \ll np$). When $r$ is substantially smaller than $np$, the conditioning event $\{\text{rank}(D) \geq r\}$ has probability close to 1, so $(r, p)$-gates behave similarly to i.i.d. $p$-gates. In this regime, the normalizing denominator in the expression for $\gamma_i$ is close to 1, yielding the approximation

$$\gamma_i \approx \log(\sqrt{2}\sigma) + \frac{1}{2}\sum_{t=r}^{n}\binom{n}{t}p^t(1-p)^{n-t}\psi\left(\frac{t-i+1}{2}\right),$$

which is the expression used in our numerical comparisons.

From $\gamma_i > \gamma_{i+1}$ we find the following corollary.

**Corollary 5.6.** *Let $1 \leq r \leq n$ be an integer, $0 < p < 1$, and $\sigma > 0$. Consider a sequence of i.i.d. $(r, p, \sigma)$-layers $(M_i)$, and write $M_i = D_i W_i$. For any positive integer $L$, define the random matrix*

$$J_L = M_L \ldots M_1 = D_L W_L D_{L-1} \ldots W_2 D_1 W_1.$$

*Denote $s_{1,L} \geq \cdots \geq s_{n,L}$ the singular values of $J_L$. Then for any $1 \leq i < r$, almost surely $s_{i,L} \neq 0$ and*

$$\frac{s_{i+1,L}}{s_{i,L}} \xrightarrow{L \to \infty} 0.$$

Our results indicate that Jacobians of FGLNs at initialization exhibit both spectral separation and depth scaling. This is confirmed by our experiments in subsection 8.

---

[1]We thank the anonymous ICML reviewer for observing that the general formula is a superposition of Lyapunov exponents of non-masked networks of varying size $t$.

# 6. Alignment of Singular Vectors Through Spectral Separation

Throughout this section, $n \geq 1$ denotes a fixed integer. We consider $n \times n$ real matrices unless specified otherwise. The goal is to formalize the following phenomenon. If a matrix $A$ has strong spectral separation, then its dominant left singular directions are stable under right-multiplication: for a wide class of matrices $B$, the product $AB$ has left singular vectors that are close to those of $A$ on the top singular subspace. Geometrically, $A$ maps the unit sphere to a highly elongated ellipsoid, and such an extreme anisotropy suppresses the influence of subsequent moderate distortions induced by $B$. We state a quantitative alignment theorem with explicit first order convergence rates (Appendix D and Remark 6.4).

Consider a product $J_L = M_L \cdots M_1$ as in fixed-gates Jacobians, and fix $\ell \ll L$. Write

$$A_\ell := M_L \cdots M_{\ell+1}, \qquad B_\ell := M_\ell \cdots M_1,$$

so that $J_L = A_\ell B_\ell$.

In the i.i.d. initialization setting, each suffix product $A_\ell$ has the same distribution as a shorter product of $(r, p, \sigma)$-layers, and thus inherits spectral separation properties analogous to Corollary 5.6. Theorem 6.1 below then predicts that the dominant left singular subspace of $J_L$ is close to that of $A_\ell$, allowing us to relate singular directions of intermediate Jacobian products to those of the full Jacobian. This is confirmed by our experiments in subsection 8.

**Theorem 6.1.** *Let $(A_L)$ be a sequence of invertible matrices such that each $A_L$ admits an SVD of the form*

$$A_L = U_{A_L} S_{A_L} I^\top,$$

*that is, the right singular vectors are the standard basis. Let $B$ be a fixed matrix and set $J_L := A_L B$, with SVD*

$$J_L = U_{J_L} S_{J_L} V_{J_L}^\top,$$

*chosen so that the $i$-th columns of $U_{A_L}$ and $U_{J_L}$ have positive dot product. Assume there exists an integer $1 \leq r \leq n - 1$ such that:*

*(i) (Spectral separation) Let $s_{i,A_L}$ be the diagonal values of $S_{A_L}$. Then*

$$\forall 1 \leq i \leq r, \qquad \frac{s_{i+1,A_L}}{s_{i,A_L}} \xrightarrow[L\to\infty]{} 0.$$

*(ii) The first $r$ columns of $B$ are linearly independent.*

*Let $R_L := U_{A_L}^\top U_{J_L}$. Write $X^{r,r}$ for the upper-left $r \times r$ block of a matrix $X$. Then*

$$R_L^{r,r} \to I_r$$

*and the off-diagonal entries converge to $0$ at rates controlled by the spectral ratios in (i). In particular, the (left) singular vectors of $A_L$ and $J_L$ align as $L \to \infty$.*

*Remark* 6.2. If $A_L = U_{A_L} S_{A_L} V_{A_L}^\top$ not assuming $V_{A_L} = I$, then the result still holds if (ii) is replaced with

(ii') The first $r$ columns of $V_{A_L}^\top B$ are linearly independent for all $L$.

If $B$ has rank $\geq r$ and the $V_L$ are Haar-distributed, this holds almost surely.

*Remark* 6.3 (Clustered spectra). When applying Theorem 6.1 to random products, it may occur that several consecutive singular values of $A_L$ are close, weakening spectral separation within that cluster. A natural extension is to assume separation only between clusters. One then expects $R_L^{r,r}$ to converge to a block-diagonal matrix with orthogonal blocks matching the cluster sizes.

*Remark* 6.4 (First order corrections). We are able to provide the first order term in the convergence $R_L^{r,r} \to I_r$ of Theorem 6.1, which we will use in Proposition 7.2. Let $C := BB^\top$. Then, as $L \to \infty$,

$$\left(S_{J_L} S_{A_L}^{-1}\right)^{r,r} \to \Sigma_C,$$
$$\left(S_{A_L}^{r,r}\right)^{-1} R_L^{r,r} S_{A_L}^{r,r} \to T_C,$$
$$S_{A_L}^{r,r} R_L^{r,r} \left(S_{A_L}^{r,r}\right)^{-1} \to (T_C^{-1})^\top,$$

where $\Sigma_C$ is diagonal and $T_C$ is lower-triangular with unit diagonal, defined by the Cholesky factorization

$$C^{r,r} = T_C \Sigma_C T_C^\top.$$

Unraveling the definitions, this means that the off-diagonal coefficient of $R_L^{r,r}$ in position $(i,j)$ decays as $\frac{s_{i,A_L}}{s_{j,A_L}}$ (resp. $\frac{s_{j,A_L}}{s_{i,A_L}}$) if $j < i$ (resp. $i < j$). The proportionality constants are related to $T_C$.

# 7. Singular Values Dynamics

Theorems 5.3 and 6.1 suggest two testable signatures of deep gated products: depth-induced scaling of ordered singular values and separation-induced alignment of dominant singular directions. In this section we illustrate how, under a controlled approximation regime combining these signatures, one can recover a deep-linear-like mode-wise singular-value evolution without assuming balancing. Consider an FGLN

$$J = W_L D_{L-1} W_{L-1} \cdots D_1 W_1$$

trained by gradient flow on the weights $(W_\ell(t))_{\ell=1}^L$ with loss $\mathcal{L}(J)$. We denote the SVD of $J = USV^\top$, with $s_1 \geq \cdots \geq s_n$ its singular values. Let

$$A_\ell := W_L D_{L-1} W_{L-1} \cdots W_{\ell+1} D_\ell,$$
$$B_\ell := D_{\ell-1} W_{\ell-1} \cdots D_1 W_1.$$

so that for all $\ell$ we have $J = A_\ell W_\ell B_\ell$. Consider their respective SVDs

$$A_\ell = U_{A_\ell} S_{A_\ell} V_{A_\ell}^\top, \qquad B_\ell = U_{B_\ell} S_{B_\ell} V_{B_\ell}^\top.$$

A derivation in Appendix E yields the mode-wise dynamics summarized below.

**Proposition 7.1.** *With notations as above, we make the following assumptions:*

*(i)* (Depth Scaling) *For each $1 \leq k \leq n$ there exists scalar functions $\gamma_k(t)$ and $\delta_k(t)$ such that for every $\ell \in \{1, \dots, L\}$ the singular values satisfy*

$$s_{k,A_\ell}(t) = e^{(L-\ell)\gamma_k(t)+\delta_k(t)},$$
$$s_{k,B_\ell}(t) = e^{((\ell-1)\gamma_k(t)+\delta_k(t)};$$

*(ii)* (Spectral separation) *There exists some $\varepsilon \ll 1$ such that for all time $t$ and every $\ell \in \{1, \dots, L\}$*

$$\|U^\top U_{A_\ell} - I\|_\infty < \varepsilon,$$
$$\|V^\top V_{B_\ell} - I\|_\infty < \varepsilon.$$

*Then for each $1 \leq k \leq n$*

$$\dot{s}_k(t) \overset{\varepsilon \to 0}{\sim} -e^{(2+\frac{2}{L})\delta_k(t)} L s_k(t)^{2-\frac{2}{L}} \langle \nabla_J \mathcal{L}(t), u_k(t)v_k(t)^\top \rangle.$$

Assumption (i) matches 0-depth scaling in Definition 4.6; it is motivated by Theorem 5.3 at initialization and serves here as a training-time approximation regime that can be tested empirically. Assumption (ii) formalizes the emergence of an approximately shared singular basis across intermediate products, as suggested by Theorem 6.1 under strong separation. Making (ii) depend only on singular values would require controlling additional quantities (e.g., Cholesky factors arising from $B_\ell B_\ell^\top$ in the alignment expansion), which we leave for future work.

Up to the multiplicative factor $e^{(2+\frac{2}{L})\delta_k(t)}$, the approximation for $\dot{s}_k$ coincides exactly with the expression stated in Proposition 4.5. In our analysis, the balancing hypothesis has been replaced by an assumption on the exponential spectra of the intermediate Jacobian whose validity during training can in principle be assessed empirically (see for instance subsections 8 and 8). As a trade-off we find an asymptotic approximation rather than an exact equality.

We emphasize that Proposition 7.1 is an illustrative derivation: its assumptions are deliberately strong and are not expected to hold uniformly across all $\ell$ (e.g., for very short prefix/suffix products). However, the statement holds for any FGLN: the gates could have arbitrary patterns not limited to the masked case. In practice, separation and alignment emerge only once products are sufficiently deep, and

alignment is typically approximate and structured (mass concentrates near the diagonal rather than collapsing exactly to $I$). Our experiments therefore evaluate these assumptions quantitatively and identify the regimes where the approximation is accurate.

In an attempt to mitigate the flaws of Proposition 7.1 (crude assumptions and restrictive model), we provide a second statement. Consider $M_1(t), \dots, M_L(t)$ any matrix-valued smooth functions, and define $J(t) = M_L(t) \dots M_1(t) = M_{1:L}(t)$, which can be considered as an MLP Jacobian, or as an intermediate Jacobian for more sophisticated architecture such as a transformer; we impose no further constraint on the time dynamics. Working in this broader setting, we may replace the order-zero spectral separation approximation in Proposition 7.1 by its first order provided by Remark 6.4.

**Proposition 7.2.** *Let $USV^\top$ be the SVD of $J$, and $s_1 \geq \dots \geq s_n$ its singular values. For each $\ell \in \{1, \dots, L\}$, we define $A_\ell(t) = M_{\ell+1:L}(t)$, $B_\ell(t) = M_{1:\ell-1}(t)$ and let their respective SVDs*

$$A_\ell = U_{A_\ell} S_{A_\ell} V_{A_\ell}^\top, \qquad B_\ell = U_{B_\ell} S_{B_\ell} V_{B_\ell}^\top.$$

*We make the following assumptions:*

*(i)* (Depth Scaling) *For each $1 \leq k \leq n$ there exists scalar functions $\gamma_k(t)$ and $\delta_k(t)$ such that the singular values satisfy*

$$s_{A_\ell,k}(t) = e^{(L-\ell)\gamma_k(t)+\delta_k(t)},$$
$$s_{B_\ell,k}(t) = e^{(\ell-1)\gamma_k(t)+\delta_k(t)};$$

*(ii)* (spectral separation) *There exists some $\varepsilon \ll 1$, lower triangular matrices $T_\ell^-(t)$ and upper triangular matrices $T_\ell^+(t)$ such that for all time $t$*

$$\|S_{A_\ell}^{-1} U^\top U_{A_\ell} S_{A_\ell} - T_\ell^-\|_\infty < \varepsilon,$$
$$\|S_{B_\ell} V_{B_\ell} V^\top S_{B_\ell}^{-1} - T_\ell^+\|_\infty < \varepsilon.$$

*Then for each $1 \leq k \leq n$*

$$\dot{s}_k \overset{\varepsilon \to 0}{\sim} e^{(1+\frac{1}{L})\delta_k} s_k^{1-\frac{1}{L}} \sum_{\ell=1}^{L} (T_\ell^- V_{A_\ell}^\top \dot{M}_\ell U_{B_\ell} T_\ell^+)_{k,k}.$$

The proof is deferred to Appendix E. At this level of generality, the resulting expression should be viewed primarily as a diagnostic tool; for a concrete architecture, one can expand $\dot{M}_\ell$ and identify the terms that dominate the singular-value dynamics. Experiments in Appendix G.3 study elementary cases.

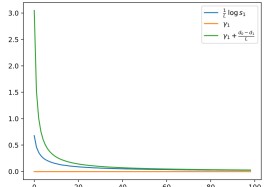 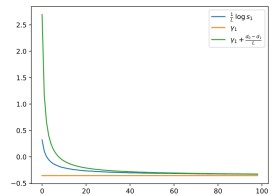

*Figure 3.* Convergence of $\frac{1}{L} \log s_{1,L}$ to $\gamma_1$ and comparison to the first-order correction $\gamma_1 + \frac{d_0 - d_1}{L}$, for Gaussian weights and Bernoulli gates with $p = 1$ (left) and $p = 0.5$ (right).

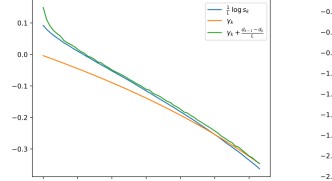 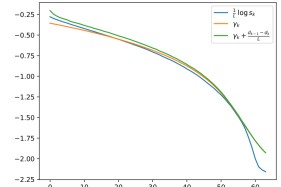

*Figure 4.* Top 64 values of $\frac{1}{L} \log s_{k,L}$ compared to $\gamma_k$ and to the corrected prediction $\gamma_k + \frac{d_{k-1} - d_k}{L}$, for depth $L = 20$ with Gaussian weights and Bernoulli gates with $p = 1$ (left) and $p = 0.5$ (right).

## 8. Experiments

We empirically evaluate the two signatures underpinning our analysis: (i) depth-induced scaling of ordered singular values at initialization (including the finite-depth correction), and (ii) separation-induced alignment of dominant singular directions in matrix products. Additional experiments and parameter sweeps are reported in Appendix G. All experiments are fully reproducible from the released implementation and scripts.[2].

**Experimental setting** We consider a square FGLN of depth $L$ and width $n = 128$ with *unconditioned* $p$-gates (i.i.d. Bernoulli$(p)$ on the diagonal) and $\sigma$-Ginibre initialized weights. Here we consider either $p = 1$ or $p = 0.5$ and always $\sigma = \frac{1}{\sqrt{n}}$. The initialization depth $L$ is specified for each plot. We used the cross entropy loss with a classic SGD optimizer.

We train the FGLN on the MNIST dataset [3] augmented with the AutoAugment policy for CIFAR-10 from Torchvision [4]. , in which case we fix parameters $L = 10$. Additional first and last layers are added to the FGLN so that the input (size 784) and output (size 10) map seamlessly to the data while still preserving its square structure for the hidden layers. For consistency with the square-matrix theory, we report spectra and alignment statistics computed on the hidden square block (the product of the $n \times n$ layers), excluding the input/output adapters. We use notations $J_L, U_{J_L}, A_\ell, U_{A_\ell}, B_\ell$ defined as in Section 7, as well as $s_{i,L}, \gamma_i, d_i$ as in Theorem 5.3. We refer to $J_L$ as the full Jacobian, and to $A_\ell, B_\ell$ as the intermediate Jacobians.

**Depth scaling** We first compare the approximation of $\frac{1}{L} \log(s_{i,L})$ provided by Theorem 5.3 with its experimental

[2]Our implementation and scripts are publicly available at https://github.com/Naloween/separation_deep_jacobian-spectra.

[3]Y. LeCun and C. Cortes. "MNIST handwritten digit database." http://yann.lecun.com/exdb/mnist/

[4]Torchvision AutoAugment implementation: https://pytorch.org/vision/stable/transforms.html#autoaugment

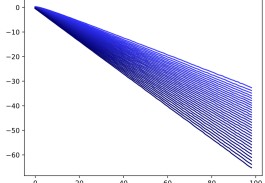 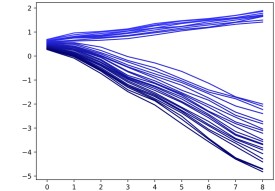

*Figure 5.* Spectrum of $\log s_{k,\ell}$ for the top 30 log-singular values of the intermediate Jacobians $B_\ell$ as $\ell$ varies, at initialization (left, $L = 100$) and trained in the MNIST setting (right), $p = 0.5$.

value at initialization in Figure 3 and Figure 4. We plot in Figure 5 the depth scaling property of the intermediate Jacobians $B_\ell$, at initialization and after training. As expected, the log-singular values are affine functions in the depth. After training, the emergence of two clusters is consistent with an effectively low-rank Jacobian geometry. This is expected to some extent: since the network output is 10 dimensions, one naturally anticipates the top 10 modes to separate from the remaining directions. The more interesting observation is that, after a fixed amount of training, this separation becomes increasingly pronounced with depth.

**Spectral separation-induced alignment** We show the alignment between the (left) singular vectors of the full Jacobian $J_L$ and the intermediate Jacobian $A_\ell$. As a metric, we use the diagonal correlation coefficient, which we recall in Appendix F. We first compare qualitatively in Figure 6 the convergence of $U_{J_L}^\top U_{A_\ell}$ to $I$ predicted by Theorem 6.1 with the experiment at initialization. In Figure 7 we compute the diagonal correlation of the matrix $U_{J_L}^\top U_{A_\ell}$. High correlation indicates that the coefficients are concentrated around the diagonal, which implies that $U_{J_L}$ and $U_{A_\ell}$ have similar columns (up to sign), that is, the (left) singular vectors are approximately aligned. At initialization the alignment is most noticeable for low values of $\ell$ as expected, while this effect is amplified and spread out to higher values of $\ell$ after training.



*Figure 6.* The matrix $U_{J_L}^\top U_{A_\ell}$ at initialization with $p = 1$. From left to right: $L = 2$, $\ell = 1$, diagonal correlation of $0.26$; $L = 10$, $\ell = 5$, diagonal correlation of $0.49$ and finally $L = 20$, $\ell = 10$, diagonal correlation of $0.84$. Note that only the upper left $64 \times 64$ sub-matrix is displayed from the original $128 \times 128$; the remaining coefficients compute the irrelevant singular vectors beyond the rank of the matrix. The fuzziness on the bottom right part of the right-most figure illustrates this effect: singular vectors in the null space of the matrix are random.

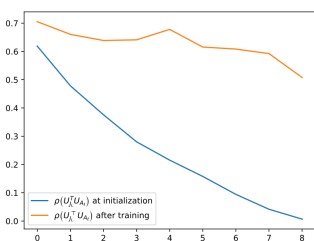

*Figure 7.* Diagonal correlation coefficient of $U_{J_L}^\top U_{A_\ell}$ at initialization (blue) and trained in the MNIST setting (orange) with $p = 0.5$

## 9. Discussion and Limitations

Our analysis identifies depth scaling and spectral-separation-induced alignment as relevant mechanisms for understanding the Jacobians of neural networks. These properties arise naturally at initialization, and our experiments suggest that they can persist, or even become more pronounced, during training. Under these assumptions, Proposition 7.1 asymptotically recovers the singular-value dynamics of balanced linear networks. This suggests that the implicit bias formally established in the balanced linear setting may extend beyond linear architectures, provided that depth scaling and spectral separation are appropriately accounted for.

Our theoretical results are mainly established for single-mode FGLNs. Figure 2 provides evidence that a related phenomenon also appears in a bias-free ReLU MLP, but extending the analysis to genuinely nonlinear multi-mode architectures remains open. Proposition 7.2 is more general, since it applies to arbitrary factored architectures and training dynamics, independently of the specific weight-update rule. However, in this level of generality, the expression is primarily diagnostic: without specifying the architecture and the dynamics of $\dot{M}_\ell$, it is difficult to determine whether training amplifies or compensates the spectral-separation effect. In the FGLN setting studied here, the effect is amplified rather than compensated.

A useful next step is to refine Proposition 7.1 under assumptions that more closely match the empirical behavior. While the full spectral-separation hypothesis is asymptotic in depth, the concentration of nonzero coefficients around the diagonal of $U_{J_L}^\top U_{A_\ell}$ appears empirically robust, as discussed in Remark 6.4. This finer alignment structure could lead to more realistic singular-value dynamics while preserving analytical tractability. Another natural direction is to analyze simple nonlinear models, such as multi-mode FGLNs with controlled dependencies between modes.

## 10. Conclusion

This paper studies how depth shapes the singular-value dynamics of neural-network Jacobians. We show that depth scaling and spectral separation can induce an alignment structure that recovers the implicit-bias dynamics of balanced linear networks in an appropriate asymptotic regime. Beyond the linear case, our experiments indicate that similar spectral organization can arise in fixed-gate and bias-free ReLU networks. These results support the view that depth does not merely rescale Jacobian dynamics, but can also structure the geometry through which gradient-based training acts.

## Impact Statement

This paper presents theoretical and empirical work aimed at improving our understanding of the optimization dynamics and implicit biases of neural networks. The contributions are primarily foundational and diagnostic, focusing on Jacobian geometry, depth scaling, and spectral separation. As such, we do not anticipate direct societal impacts beyond those associated with progress in machine learning research more broadly.

## Acknowledgments

This work was supported by ANR-22-CE23-0002 ERIANA and ANR-22-EXES-0009 MAIA, by the CNRS MITI interdisciplinary programs through the PRIME exploratory research program (PRIME CNRS AIM-GPT), and by access to the HPC resources of IDRIS under GENCI allocation AD011013338.

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

# A. Proof of weak singular value dynamics on Fixed Gates Linear Networks

**Lemma A.1.** *Consider an FGLN $J(t) = W_L(t)D_{L-1} \cdots D_1 W_1(t)$ trained by gradient flow on the weights $(W_\ell(t))_{\ell=1}^L$ with loss $\mathcal{L}(J)$. Define $M_\ell(t) := D_\ell W_\ell(t) D_{\ell-1}$ for $\ell = 1, \ldots, L$. Then:*

$$\nabla_{W_\ell}\mathcal{L} = D_\ell M_{\ell+1:L}^\top \nabla_J \mathcal{L}(J) M_{1:\ell-1}^\top$$

*Proof.* Let $A = M_{\ell+1:L} D_\ell$ and $B = M_{1:\ell-1}$ We write the network as

$$J(W_\ell) = AW_\ell B.$$

Denote $\langle X, Y \rangle = \text{tr}(X^T Y)$ the Frobenius inner product. Let $H$ be any matrix. The chain rule gives

$$\langle \nabla_{W_\ell}\mathcal{L}(J), H \rangle = \langle \nabla_J \mathcal{L}(J), AHB \rangle = \langle A^\top \nabla_J \mathcal{L}(J) B^\top, H \rangle.$$

Since this holds for all $H$, the equality follows. □

We may now prove Proposition 4.5.

*Proof of proposition 4.5.* We define the difference:

$$\Delta_\ell(t) := M_{\ell+1}^\top(t) M_{\ell+1}(t) - M_\ell(t) M_\ell^\top(t)$$

We then show $\dot{\Delta}_\ell(t) = 0$, which implies $\Delta_\ell(t) = 0$ for all $t$. Indeed, observe that

$$\dot{M}_\ell = D_\ell \dot{W}_\ell D_{\ell-1}$$

Develop $\dot{\Delta}_\ell$:

$$\dot{\Delta}_\ell = \dot{M}_{\ell+1}^\top M_{\ell+1} + M_{\ell+1}^\top \dot{M}_{\ell+1} - \left( \dot{M}_\ell M_\ell^\top + M_\ell \dot{M}_\ell^\top \right) \tag{1}$$

Each factors can be expanded with the help of Lemma A.1:

$$\begin{aligned}
\dot{M}_{\ell+1}^\top M_{\ell+1} &= -D_\ell \left( D_{\ell+1} M_{\ell+2:L}^\top \nabla_J \mathcal{L} \, M_{1:\ell}^\top \right)^\top D_{\ell+1} D_{\ell+1} W_{\ell+1} D_\ell \\
&= -D_\ell M_{1:\ell} \nabla_J \mathcal{L}^\top M_{\ell+2:L} D_{\ell+1} W_{\ell+1} D_\ell \\
&= -M_{1:\ell} \nabla_J \mathcal{L}^\top M_{\ell+1:L}
\end{aligned}$$

Similarly,

$$\begin{aligned}
M_\ell \dot{M}_\ell^\top &= -M_{1:\ell} \nabla_J \mathcal{L}^\top M_{\ell+1:L} \\
M_{\ell+1}^\top \dot{M}_{\ell+1} &= -M_{\ell+1:L}^\top \nabla_J \mathcal{L} M_{1:\ell}^\top \\
\dot{M}_\ell M_\ell^\top &= -M_{\ell+1:L}^\top \nabla_J \mathcal{L} M_{1:\ell}^\top.
\end{aligned}$$

We see that all terms in equation 1 cancel out. The rest is identical to the proof done by (Arora et al., 2019) in the deep linear case. □

# B. Recollections on exterior powers

In the course of the proof of Theorems 5.3 and 6.1 in Appendices C and D we rely on several occasions on the notion of exterior powers. We collect here their definition and fundamental and properties for the reader's convenience.

From now-on fix $V$ a real vector space of dimension $n \geq 1$, which can be thought of as $\mathbb{R}^n$. Let $t \geq 1$ be an integer.

## B.1. Tensor powers

Tensor powers are more common and easier to grasp than exterior powers. Although we do not use them in our proofs, we introduce them now to draw parallels later on.

### B.1.1. DESCRIPTION OF TENSOR POWERS AND TENSORS

**Definition B.1.** The *(t-fold) tensor power* (or *tensor product*) of $V$ is the vector space denoted $\bigotimes^t V$ generated by symbols

$$v_1 \otimes \cdots \otimes v_t, \qquad v_1, \ldots, v_t \in V$$

subject to the following multi-linear relations: for every $1 \leq i \leq t$, $x_i, y_i, v_1, \ldots, v_t \in V$, $a \in \mathbb{R}$,

$$v_1 \otimes \cdots \otimes (ax_i + y_i) \otimes \cdots \otimes v_t = a(v_1 \otimes \cdots \otimes x_i \otimes \cdots \otimes v_t) + v_1 \otimes \cdots \otimes y_i \otimes \cdots \otimes v_t.$$

Vectors of $\bigotimes^t V$ are often called *tensors*.

Naturally, $\bigotimes^1 V = V$. The tensor product $\bigotimes^t V$ is *generated* as a vector space by the tensors $v_1 \otimes \cdots \otimes v_t$ called *elementary tensors*. This implies that a general tensor in $\bigotimes^t V$ is a sum

$$\sum_i v_{1,i} \otimes \cdots \otimes v_{t,i}.$$

Most tensors in $\bigotimes^t V$ cannot be simplified to an elementary tensor. For instance with $V = \mathbb{R}^2$ with standard basis $(e_1, e_2)$, $t = 2$, the tensor $e_1 \otimes e_1 + e_2 \otimes e_2$ is not elementary.

The $\otimes$ symbol is not commutative: for $v, w \in V$ and $t = 2$, the tensors $v \otimes w$ and $w \otimes v$ in $\bigotimes^2 V$ are different, unless $v = w$.

Fix a basis $(e_1, \ldots, e_n)$ on $V$. Then $\bigotimes^t V$ is naturally equipped with a basis composed of the elementary tensors of the form

$$e_I := e_{i_1} \otimes \cdots \otimes e_{i_t}, \qquad I = (i_1, \ldots, i_t) \subseteq \{1, \ldots, n\}^t.$$

In particular, $\bigotimes^t V$ has dimension $n^t$. We may order the basis tensors $(e_I)$ using the lexicographic ordering.

If $V$ is moreover equipped with a dot product $\langle, \rangle$, then so is $\bigotimes^t V$. It suffices to specify the dot product on elementary tensors, for which we set

$$\langle v_1 \otimes \cdots \otimes v_t, w_1 \otimes \cdots \otimes w_t \rangle := \prod_{i=1}^{t} \langle v_i, w_i \rangle.$$

If a basis $(e_1, \ldots, e_n)$ is orthonormal with respect to $\langle, \rangle$, then so is the induced basis $(e_I)$ on $\bigotimes^t V$ with respect to the induced dot product.

### B.1.2. TENSOR POWERS OF MATRICES

Let $f$ be an endomorphism of $V$. Then we can define an endomorphism $\otimes^t f$ acting on $V$ by setting for every $v_1, \ldots, v_t \in V$

$$(\otimes^t f)(v_1 \otimes \cdots \otimes v_t) = f(v_1) \otimes \cdots \otimes f(v_t).$$

By specifying a basis, we can thus make sense of the tensor powers of a matrix.

**Definition B.2.** Let $(e_i) = (e_1, \ldots, e_n)$ the standard basis for $\mathbb{R}^n$, consider $(e_I)$ the induced basis on $\bigotimes^t \mathbb{R}^n$ ordered as above. Let $M$ be an $n \times n$ matrix, denote $f$ the automorphism of $\mathbb{R}^n$ it represents in the basis $(e_i)$. Then $\otimes^t M$ is the $n^t \times n^t$ matrix representing $\otimes^t f$ in the basis $(e_I)$ of $\bigotimes^t \mathbb{R}^n$.

If $M = (m_{i,j})$, then for $I, J \subseteq \{1, \ldots, n\}^t$, $I = (i_1, \ldots, i_t)$, $J = (j_1, \ldots, j_t)$, the coefficient in position $(I, J)$ of $\otimes^t M$ is

$$(\otimes^t M)_{I,J} = \prod_{k=1}^t m_{i_k, j_k}.$$

Tensor powers are compatible with matrix multiplication:

$$\otimes^t(MN) = (\otimes^t M)(\otimes^t N).$$

If $u_1, \ldots, u_n$ denote the singular vectors of $M$ in $\mathbb{R}^n$, with corresponding singular values $s_1 \geq \cdots \geq s_n$, then the singular vectors of $\otimes^t M$ are exactly the elementary tensors

$$u_{i_1} \otimes \cdots \otimes u_{i_t}, \qquad I = (i_1, \ldots, i_t) \subseteq \{1, \ldots, n\}^t$$

with corresponding singular values given by the product $\prod_{i \in I} s_i$. In particular, (one of) the top singular vector(s) of $\otimes^t M$ is $u_1 \otimes \cdots \otimes u_1$ with singular value $s_1^t$.

## B.2. Exterior powers

We define and describe properties of exterior powers. The first two paragraphs are parallel to the ones on tensor powers. The third paragraph recalls the geometric meaning associated with exterior powers.

### B.2.1. DESCRIPTION OF EXTERIOR POWERS AND WEDGES

**Definition B.3.** The *(t-fold) exterior power* (or *wedge product*) of $V$ is the vector space denoted $\bigwedge^t V$ generated by symbols

$$v_1 \wedge \cdots \wedge v_t, \qquad v_1, \ldots, v_t \in V$$

subject to the same multi-linear relations as in Definition B.1, as well as the additional skew-symmetry relation: for every permutation of $\{1, \ldots, n\}$ denoted $\sigma$, for every $v_1, \ldots, v_t \in V$,

$$v_{\sigma(1)} \wedge \cdots \wedge v_{\sigma(t)} = \varepsilon(\sigma)(v_1 \wedge \cdots \wedge v_t)$$

where $\varepsilon(\sigma)$ denotes the signature of the permutation $\sigma$. Vectors of $\bigwedge^t V$ are often called *wedges*.

Naturally, $\bigwedge^1 V = V$. The wedge product $\bigwedge^t V$ is *generated* as a vector space by the wedges $v_1 \wedge \cdots \wedge v_t$ called *elementary wedges*. This implies that a general wedge in $\bigwedge^t V$ is a sum

$$\sum_i v_{1,i} \wedge \cdots \wedge v_{t,i}.$$

Most wedges in $\bigwedge^t V$ cannot be simplified to an elementary tensor. For instance with $V = \mathbb{R}^4$ with standard basis $(e_1, e_2, e_3, e_4)$, $t = 2$, the wedge $e_1 \wedge e_2 + e_3 \wedge e_4$ is not elementary.

The $\wedge$ symbol is skew-commutative: for $v, w \in V$ and $t = 2$, we have $v \wedge w = -w \wedge v$. Hence if $v = w$ then $v \wedge w = 0$. For a $t$-fold wedge $v_1 \wedge \cdots \wedge v_t$, swapping any two components induces a change of sign.

Fix a basis $(e_1, \ldots, e_n)$ on $V$. Then $\bigwedge^t V$ is naturally equipped with a basis composed of the elementary wedges of the form

$$e_I := e_{i_1} \wedge \cdots \wedge e_{i_t}, \qquad I = (i_1 < \cdots < i_t) \subseteq \{1, \ldots, n\}^t.$$

In particular, $\bigwedge^t V$ has dimension $\binom{n}{t}$. If $t = n$, then $\bigwedge^t V$ is one-dimensional, and for $t > n$ we find $\bigwedge^t V = \{0\}$. We may order the basis wedges $(e_I)$ using the lexicographic ordering.

If $V$ is moreover equipped with a dot product $\langle , \rangle$, then so is $\bigwedge^t V$. It suffices to specify the dot product on elementary wedges, for which we set

$$\langle v_1 \wedge \cdots \wedge v_t, w_1 \wedge \cdots \wedge w_t \rangle := \det((\langle v_i, w_j \rangle)_{i,j}).$$

If a basis $(e_1, \ldots, e_n)$ is orthonormal with respect to $\langle , \rangle$, then so is the induced basis $(e_I)$ on $\bigwedge^t V$ with respect to the induced dot product.

### B.2.2. EXTERIOR POWERS OF MATRICES

Let $f$ be an endomorphism of $V$. Then we can define an endomorphism $\wedge^t f$ acting on $V$ by setting for every $v_1, \ldots, v_t \in V$

$$(\wedge^t f)(v_1 \wedge \cdots \wedge v_t) = f(v_1) \wedge \cdots \wedge f(v_t).$$

By specifying a basis, we can thus make sense of the exterior powers of a matrix.

**Definition B.4.** Let $(e_i) = (e_1, \ldots, e_n)$ the standard basis for $\mathbb{R}^n$, consider $(e_I)$ the induced basis on $\bigwedge^t \mathbb{R}^n$ ordered as above. Let $M$ be an $n \times n$ matrix, denote $f$ the automorphism of $\mathbb{R}^n$ it represents in the basis $(e_i)$. Then $\wedge^t M$ is the $\binom{n}{t} \times \binom{n}{t}$ matrix representing $\wedge^t f$ in the basis $(e_I)$ of $\bigwedge^t \mathbb{R}^n$.

If $M = (m_{i,j})$, then for $I, J \subseteq \{1, \ldots, n\}^t$, $I = (i_1 < \cdots < i_t)$, $J = (j_1 < \cdots < j_t)$, the coefficient in position $(I, J)$ of $\wedge^t M$ is

$$(\wedge^t M)_{I,J} = \det M^{I,J}$$

where $M^{I,J}$ denotes the sub-matrix of $M$ keeping only the lines (resp. columns) indexed by $I$ (resp. $J$). Exterior power is compatible with matrix multiplication:

$$\wedge^t(MN) = (\wedge^t M)(\wedge^t N).$$

If $u_1, \ldots, u_n$ denote the singular vectors of $M$ in $\mathbb{R}^n$, with corresponding singular values $s_1, \ldots, s_n$, then the singular vectors of $\wedge^t M$ are exactly the elementary wedges

$$u_{i_1} \wedge \cdots \wedge u_{i_t}, \qquad I = (i_1 < \cdots < i_t) \subseteq \{1, \ldots, n\}^t$$

with corresponding singular values given by the product $\prod_{i \in I} s_i$. In particular, (one of) the top singular vector(s) of $\wedge^t M$ is $u_1 \wedge u_2 \wedge \cdots \wedge u_t$ with singular value $s_1 s_2 \ldots s_t$.

*Remark* B.5. This last observation is the reason why exterior powers are naturally involved as soon as one is interested in singular values of a matrix $M$ other than the top one.

### B.2.3. GEOMETRIC MEANING OF EXTERIOR POWERS

Intuitively the exterior power $\bigwedge^t V$ should be thought of as the space generated by $t$-dimensional linear subspaces of $V$. In particular, if $v_1, \ldots, v_t \in V$ are linearly independent one should think about the elementary wedge $v_1 \wedge \cdots \wedge v_t$ as representing the subspace spanned by $v_1, \ldots, v_t$.

Let $v_1, \ldots, v_t$ (resp. $w_1, \ldots, w_t$) be linearly independent vectors in $V$. Denote $E$ (resp. $F$) the subspace spanned by the $v_i$ (resp. $w_i$). Then

$$v_1 \wedge \cdots \wedge v_t = a(w_1 \wedge \cdots \wedge w_t) \text{ for some scalar } a \in \mathbb{R} \text{ if and only if } E = F.$$

If $V$ is equipped with a dot product $\langle , \rangle$ then the induced dot product on $\bigwedge^t V$ corresponds to the intuitive notion of angle between $t$-dimensional subspaces. In particular:

$$\langle v_1 \wedge \cdots \wedge v_t, w_1 \wedge \cdots \wedge w_t \rangle = 0 \text{ if and only if } E \text{ and } F \text{ are at right angle.}$$

Suppose that each $v_i = v_{i,L}$ depends on some integer $L$. Assume each $v_{i,L}$ and $w_i$ is a unit vector. Setting $E_L = \text{Span}(v_{1,L}, \ldots, v_{t,L})$, then

$$v_{1,L} \wedge \cdots \wedge v_{t,L} \xrightarrow{L \to \infty} \pm(w_1 \wedge \cdots \wedge w_t) \text{ if and only if } E_L \text{ aligns with } F \text{ as } L \to \infty.$$

# C. Proof of theorem 5.3

We divide the proof in two steps. First we show the existence and formula for the Lyapunov exponents $\gamma_i$ using standard results from the literature of products random matrices. Then we compute a second order term for the expectation $\mathbb{E}[\log s_{i,L}]$; this is achieved by studying the quotient of operator norms $\|AB\|/\|A\|\|B\|$ when $A$ and $B$ are random square matrices with spectral separation.

## C.1. First half of the proof of Theorem 5.3

For the rest of this paragraph we fix $1 \le r \le n$, $0 < p < 1$ and $\sigma > 0$. We seek to prove the existence and value for the Lyapunov exponents $\gamma_i$ in Theorem 5.3, which already implies Corollary 5.6.

We begin by stating an application of Theorem 2 from (Furstenberg & Kesten, 1960).

**Theorem C.1.** *Let $(Y_i)$ be a sequence of i.i.d. random matrices. For any positive integer L, define the random matrix*

$$J_L = Y_L \dots Y_1.$$

*Assume that*

- *The expectation $\mathbb{E}[\log^+(\|Y_1\|)]$ is finite, and*

- *almost surely, $J_L$ is nonzero for every L.*

*Denote $s_{1,L} \ge \dots \ge s_{n,L}$ the singular values of $J_L$. Then there exists numbers $\gamma_1 \ge \dots \ge \gamma_n$ (possibly $-\infty$) such that for any $1 \le i \le n$, almost surely*

$$\frac{1}{L} \log s_{i,L} = \gamma_i.$$

*Moreover, if for any orthogonal matrix $U$, $Y_1 U$ has same distribution as $Y_1$, then for any $1 \le i \le n$*

$$\gamma_1 + \dots + \gamma_i = \mathbb{E}[\log \|Y_1 e_1 \wedge \dots \wedge Y_1 e_i\|].$$

The existence part in Theorem 5.3 follows from Theorem C.1 once we check that the assumptions hold.

**Lemma C.2.** *A sequence of i.i.d. $(r, p, \sigma)$-layers $(Y_i)$ satisfies the assumptions of Theorem C.1.*

*Proof.* Write $Y_1 = D_1 W_1$. For any orthogonal matrix $U$, the product $W_1 U$ has same distribution as $W_1$, so the same holds true of $Y_1$. It is known that $\mathbb{E}[\log^+ \|W_1\|]$ is finite, thus

$$0 \le \mathbb{E}[\log^+ \|Y_1\|] \le \mathbb{E}[\log^+ \|D_1\| \cdot \|W_1\|] = \mathbb{E}[\log^+ \|W_1\|] < \infty.$$

We show that $J_L$ is nonzero for every $L$ almost surely by showing inductively that it has rank at least $r$. This is clearly the case of $J_1$. If $J_L$ has rank $\ge r$, then so does $W_{L+1} J_L$ almost surely. The rank of $J_{L+1} = D_{L+1} W_{L+1} J_L$ is given by

$$\mathrm{rk}(D_{L+1}) - \dim(\mathrm{Im}(W_{L+1} J_L) \cap \ker D_{L+1})$$

Recall that the dimension of the intersection of independent uniformly distributed subspaces of dimensions $d_1$ and $d_2$ is almost surely $\max(0, n - d_1 - d_2)$, so the dimension of $\ker J_{L+1}$ is almost surely

$$\mathrm{rk}(J_{L+1}) = \mathrm{rk}(D_{L+1}) - \max(0, -\mathrm{rk}(J_L) + \mathrm{rk}(D_{L+1})) = \min(\mathrm{rk}(D_{L+1}), \mathrm{rk}(J_L)) \ge r.$$

$\square$

We now compute the Lyapunov exponents. The appearance of wedge products in the formula

$$\gamma_1 + \dots + \gamma_i = \mathbb{E}[\log \|Y_1 e_1 \wedge \dots \wedge Y_1 e_i\|]$$

given in Theorem C.1 is explained by Remark B.5. We compute this expectation in two steps.

**Lemma C.3.** *Let $W$ be a $\sigma$-Ginibre matrix. For $1 \leq t \leq n$, let $D$ be diagonal with entries in $\{0, 1\}$, of rank $t$ (i.e. there are $t$ copies of $1$ along the diagonal). Set $Y = DW$. Then for any $1 \leq i \leq n$,*

$$\mathbb{E}[\log \|Ye_1 \wedge \cdots \wedge Ye_i\|] = \begin{cases} \frac{i}{2}\log(2\sigma^2) + \frac{1}{2}\sum_{k=1}^{i}\psi\left(\frac{t-k+1}{2}\right) & \text{if } i \leq r \\ -\infty & \text{if } i > r \end{cases}$$

*where $\psi$ denotes the digamma function.*

*Proof.* Up to relabeling, we may assume that

$$D = \text{diag}(1, \ldots, 1, 0, \ldots, 0).$$

Observe that $We_j$ is the $j$-th column of $W$, hence $Ye_j$ is a vector whose first $t$ coordinates coincide with those of $We_j$, and the rest are zeros. Thus we can view $(Ye_j)_{1 \leq j \leq i}$ as a collection of independent Gaussian vectors in $\mathbb{R}^t$. If $i > t$, this collection is not linearly independent, hence

$$Ye_1 \wedge \cdots \wedge Ye_i = 0.$$

From now on we assume $i \leq t$. Denoting $\langle, \rangle$ the standard scalar product in $\mathbb{R}^t$, the norm on the wedge product $\bigwedge^i \mathbb{R}^t$ satisfies (see paragraph B.2.1)

$$\|Ye_1 \wedge \cdots \wedge Ye_i\|^2 = \det(\langle Ye_p, Ye_q \rangle)_{1 \leq p,q \leq i} = \det(X^\top X)$$

where $X$ is the $t \times i$ matrix whose columns are the independent Gaussian vectors $Ye_1, \ldots, Ye_i$ viewed in $\mathbb{R}^t$. The $i \times i$ matrix $X^\top X$ thus follows the Wishart distribution, it is known that the log-expectation of its determinant is

$$\mathbb{E}[\log \det(X^\top X)] = i\log(2\sigma^2) + \sum_{k=1}^{i}\psi\left(\frac{t-k+1}{2}\right).$$

$\square$

**Lemma C.4.** *Let $Y_1 = D_1 W_1$ be an $(r, p, \sigma)$-layer. For any $1 \leq i \leq n$, denote $\lambda_i = \mathbb{E}[\log \|Y_1 e_1 \wedge \cdots \wedge Y_1 e_i\|]$. Then*

$$\lambda_i = \begin{cases} -\infty & \text{if } i > r \\ \frac{i}{2}\log(2\sigma^2) + \frac{\sum_{t=r}^{n}\left[\binom{n}{t}p^t(1-p)^{n-t}\sum_{k=1}^{i}\psi\left(\frac{t-k+1}{2}\right)\right]}{2\sum_{m=r}^{n}\binom{n}{m}p^m(1-p)^{n-m}} & \text{otherwise.} \end{cases}$$

*where $\psi$ denotes the digamma function. For $i \leq r \ll np$ the expression simplifies to*

$$\lambda_i \approx \frac{i}{2}\log(2\sigma^2) + \frac{1}{2}\sum_{t=r}^{n}\left[\binom{n}{t}p^t(1-p)^{n-t}\sum_{k=1}^{i}\psi\left(\frac{t-k+1}{2}\right)\right]$$

*Proof.* We split the expectation according to the rank of $D_1$. If $i > r$, some terms are $-\infty$, hence $\lambda_i = -\infty$. From now-on we assume $i \leq r$.

By definition for $r \leq t \leq n$

$$P(\text{rk}D_1 = t) = \frac{P(\text{rk}D = t)}{P(\text{rk}D \geq r)} = \frac{\binom{n}{t}p^t(1-p)^{n-t}}{\sum_{m=r}^{n}\binom{n}{m}p^m(1-p)^{n-m}}$$

where $D$ denotes a $(p)$-gate, whose rank follows an $(n, p)$-binomial distribution. Hence, letting $\lambda_i = \mathbb{E}[\log \|Y_1 e_1 \wedge \cdots \wedge Y_1 e_i\|]$, we find

$$\lambda_i = \frac{i}{2}\log(2\sigma^2) + \frac{\sum_{t=r}^{n}\left[\binom{n}{t}p^t(1-p)^{n-t}\sum_{k=1}^{i}\psi\left(\frac{t-k+1}{2}\right)\right]}{2\sum_{m=r}^{n}\binom{n}{m}p^m(1-p)^{n-m}}$$

When $r \ll np$, the sum in the denominator is almost indistinguishable from 1, hence the approximation. $\square$

We deduce the values of the exponents $\gamma_i$.

**Lemma C.5.** *Let $Y_1 = D_1 W_1$ be an $(r, p, \sigma)$-layer. For any $1 \leq i \leq r$, denote $\lambda_i = \mathbb{E}[\log \|Y_1 e_1 \wedge \cdots \wedge Y_1 e_i\|]$ and $\gamma_i := \lambda_i - \lambda_{i-1}$. Then*

$$\gamma_i = \log(\sqrt{2}\sigma) + \frac{\sum_{t=r}^n \left[ \binom{n}{t} p^t (1-p)^{n-t} \psi\left(\frac{t-i+1}{2}\right) \right]}{2 \sum_{m=r}^n \binom{n}{m} p^m (1-p)^{n-m}}$$

$$\approx \log(\sqrt{2}\sigma) + \frac{1}{2} \sum_{t=r}^n \left[ \binom{n}{t} p^t (1-p)^{n-t} \psi\left(\frac{t-i+1}{2}\right) \right] \qquad \text{for } r \ll np.$$

*where $\psi$ denotes the digamma function. Moreover, $\gamma_i > \gamma_{i+1}$ for $i < r$.*

*Proof.* The value for $\gamma_i$ and its approximation follow from those of $\lambda_i$. If $i < r$, as $\psi$ is increasing, we find $\gamma_i > \gamma_{i+1}$. $\square$

### C.2. Second half of the proof of Theorem 5.3

Again we fix $1 \leq r \leq n$, $0 < p < 1$ and $\sigma > 0$. We wish to compute the limit of the quantity $\mathbb{E}[\log(s_{i,L})] - \gamma_i L$ as $L \to \infty$.

Let us motivate the Lemma C.6 below. If $A$ and $B$ are two square matrices, then

$$\|AB\| \leq \|A\| \|B\|.$$

In general the gap between both quantities can be arbitrary, as it heavily depends on the relative orientation of singular vectors of $A$ and $B$. However if $A$ and $B$ exhibit spectral separation and Haar-distributed singular vectors, then it is possible to estimate on average the value of the ratio $\frac{\|AB\|}{\|A\| \|B\|}$. This is a particular case of the content of Lemma C.6.

**Lemma C.6.** *Let $(A_L)_L, (B_L)_L$ be sequences of random matrices. For every $1 \leq i \leq n$, denote $s_{i,L}^A$ (resp. $s_{i,L}^B$) the $i$-th singular value of $A_L$ (resp. $B_L$), in decreasing order. Assume further that*

*(a) for $1 \leq i \leq r$, almost surely $s_{i,L}^A \neq 0$ and $\frac{s_{i+1,L}^A}{s_{i,L}^A} \xrightarrow{L \to \infty} 0$, similarly for B,*

*(b) writing $A_L = U_L^A S_L^A V_L^A$ and $B_L = U_L^B S_L^B V_L^B$ for the respective SVDs, $V_L^A U_L^B$ is Haar distributed and independent of $(S_L^A, S_L^B)$.*

*Then for every $1 \leq t \leq r$,*

$$\mathbb{E}\left[\log \frac{\|\wedge^t (A_L B_L)\|}{\|\wedge^t A_L\| \|\wedge^t B_L\|}\right] \xrightarrow{L \to \infty} \mathbb{E}[\log |\det \Omega^{t,t}|] \leq 0$$

*where $\Omega^{t,t}$ is the $t \times t$ top-left block of a Haar-distributed orthogonal matrix.*

For the notions and properties of exterior powers of matrices in the statement and proof of Lemma C.6, see paragraph B.2.2.

*Proof.* Fix some integer $L$. It is harmless to multiply $A_L$ on the left (resp. $B_L$ on the right) by any random orthogonal matrix. Thus we can assume that $A_L = S_L^A V_L^A$, $B_L = U_L^B S_L^B$. Define

$$\Omega = V_L^A U_L^B$$

which depends on $L$ (but we omit the subscript for simplicity). By definition

$$A_L B_L = S_L^A \Omega S_L^B.$$

We are interested in the log-expectation

$$\mathbb{E}\left[\log \frac{\|\wedge^t (A_L B_L)\|}{\|\wedge^t A_L\| \|\wedge^t B_L\|}\right] = \mathbb{E}\left[\log \frac{\|(\wedge^t S_L^A) \times (\wedge^t \Omega) \times (\wedge^t S_L^B)\|}{\|\wedge^t S_L^A\| \|\wedge^t S_L^B\|}\right]$$

with

$$\wedge^t S_L^A = \mathrm{diag}\left(\prod_{i\in I} s_{i,L}^A\right)_I , \quad \wedge^t \Omega = (\det \Omega^{I,J})_{I,J}, \quad \wedge^t S_L^B = \mathrm{diag}\left(\prod_{j\in J} s_{j,L}^B\right)_J .$$

Let $I_0 = \{1, \ldots, t\}$ which is the leading index for the standard basis of $\bigwedge^t \mathbb{R}^n$. Then

$$\| \wedge^t S_L^A \| = \prod_{i\in I_0} s_{i,L}^A, \quad \| \wedge^t S_L^B \| = \prod_{i\in I_0} s_{i,L}^B$$

hence, by assumption *(a)* the diagonal matrices $S_L^A/\| \wedge^t S_L^A \|$ and $S_L^B/\| \wedge^t S_L^B \|$ almost surely exist and converge to

$$E_{I_0,I_0} := \mathrm{diag}(1, 0 \ldots, 0).$$

By assumption *(b)* the matrix $\Omega$ is independent of $S_L^A$ and $S_L^B$ and is identically distributed as $L$ varies, thus we may assume that it does not depend on $L$: the log-expectation above remains unchanged for every $L$. In this setting, almost surely

$$\frac{1}{\| \wedge^t A_L \|\| \wedge^t B_L \|} \wedge^t (A_L B_L) \xrightarrow{L\to\infty} E_{I_0,I_0} \times \wedge^t \Omega \times E_{I_0,I_0}.$$

Observe that the matrix on the right has vanishing coefficients everywhere except at index $(I_0, I_0)$ where it evaluates to $\det \Omega^{I_0,I_0}$:

$$\|E_{I_0,I_0} \times \wedge^t \Omega \times E_{I_0,I_0}\| = |\det \Omega^{I_0,I_0}|.$$

We deduce the convergence of the log-expectation via the dominated convergence theorem, with domination

$$\log |\det \Omega^{I_0,I_0}| \le \log \frac{\| \wedge^t (A_L B_L)\|}{\| \wedge^t A_L \|\| \wedge^t B_L \|} \le 0.$$

where the inequality on the left follows from $\| \cdot \|_\infty \le \| \cdot \|$ as matrix norms. We have shown

$$\mathbb{E}\left[\log \frac{\| \wedge^t (A_L B_L)\|}{\| \wedge^t A_L \|\| \wedge^t B_L \|}\right] \xrightarrow{L\to\infty} \mathbb{E}[\log |\det \Omega^{I_0,I_0}|]$$

which concludes by observing that the variable $\Omega^{t,t}$ from the statement has same distribution as $\Omega^{I_0,I_0}$. $\square$

We make use of Lemma C.6 as follows. With notations of Theorem C.1 it follows from the almost sure convergence and the dominated convergence theorem that

$$\gamma_1 + \cdots + \gamma_i = \lim_{L\to\infty} \frac{1}{L}\mathbb{E}[\log(s_{1,L} \ldots s_{i,L})] = \lim_{L\to\infty} \frac{1}{L}\mathbb{E}[\log \| \wedge^i J_L \|].$$

Set $u_L = \mathbb{E}[\log \| \wedge^i J_L \|]$ and write

$$u_{p+q} + d(p, q) = u_p + u_q$$

with by definition

$$d(p, q) = -\mathbb{E}\left[\log \frac{\| \wedge^i (J_p J_q)\|}{\| \wedge^i J_p \|\| \wedge^i J_q \|}\right].$$

By Corollary 5.6 and properties of $\sigma$-Ginibre matrices the assumptions of Lemma C.6 are satisfied in the setting of Theorem 5.3, thus

$$d(p, q) \xrightarrow{(p,q)\to\infty} -\mathbb{E}[\log |\det \Omega_i|] := d_i \ge 0.$$

From Lemma C.7 below we find

$$\frac{1}{L}u_L = \gamma_1 + \cdots + \gamma_i + \frac{d_i}{L} + o\left(\frac{1}{L}\right)$$

Subtracting consecutive values of $i$, we obtain the final statement of Theorem 5.3.

**Lemma C.7.** *Let $(u_L)$ be a sequence of real numbers satisfying for all $p, q \geq 1$*

$$u_{p+q} + d(p, q) = u_p + u_q$$

*with $d(p, q) \geq 0$ and $d(p, q) \xrightarrow{(p,q)\to\infty} d$. Then the sequence $(u_L/L)$ admits a finite limit $u$, and*

$$\frac{1}{L}u_L = u + \frac{d}{L} + o\left(\frac{1}{L}\right).$$

*Proof.* Replacing $u_L$ with $u_L + (L-1)d$, we may assume $d = 0$. By Fekete's subadditive lemma, the sequence $(u_L/L)$ converges in $[-\infty, +\infty[$, let $u$ be its limit. Fix some integer $L \geq 2$, induction shows that for every $m \geq 1$,

$$\frac{1}{L^m}u_{L^m} + (L-1)\sum_{k=2}^{m} \frac{\frac{1}{L-1}\sum_{i=1}^{L-1} d(L^{k-1}, iL^{k-1})}{L^k} = \frac{1}{L}u_L.$$

We begin by showing that the limit $u$ is finite. By assumption on $d(\cdot, \cdot)$, for any integer $L$ and $2 \leq i < L$ the quantity $d(L^{k-1}, iL^{k-1})$ is bounded as a function of $k$. Hence the same can be said of the average

$$\frac{1}{L-1}\sum_{i=1}^{L-1} d(L^{k-1}, iL^{k-1})$$

which ensures the convergence of the series

$$0 \leq \sum_{k=2}^{\infty} \frac{\frac{1}{L-1}\sum_{i=1}^{L-1} d(L^{k-1}, iL^{k-1})}{L^k} < \infty$$

which in turn shows that the limit $u = \lim_{m\to\infty} \frac{1}{L^m}u_{L^m}$ is finite. We have thus established that for any $L \geq 2$,

$$\frac{1}{L}u_L = u + (L-1)\sum_{k=2}^{\infty} \frac{\frac{1}{L-1}\sum_{i=1}^{L-1} d(L^{k-1}, iL^{k-1})}{L^k}$$

Consider $\varepsilon > 0$ and $L_0$ such that for any $L \geq L_0$ and $1 \leq i < L$, $0 \leq d(L^{k-1}, iL^{k-1}) < \varepsilon$. Hence for $L \geq L_0$

$$\left|\frac{1}{L}u_L - u\right| \leq \varepsilon \sum_{k=2}^{\infty} \frac{L-1}{L^k} = \frac{\varepsilon}{L}.$$

Thus $L(\frac{1}{L}u_L - u) \xrightarrow{L\to\infty} 0$ which is what we needed to show. $\qquad\square$

## D. Proof of theorem 6.1

### D.1. Another statement

Theorem 6.1 is a consequence of the slightly more general statement.

**Theorem D.1.** *Let $(S_L)$ be a sequence of diagonal matrices and let $C = (c_{i,j})$ be a symmetric matrix such that*

(i) *The diagonal coefficients of $S_L$ are non-negative non-increasing: $s_{1,L} \geq \cdots \geq s_{n,L} \geq 0$,*

(ii) *(spectral separation) There is an integer $1 \leq r \leq n-1$ such that*

$$\forall 1 \leq i \leq r, \quad \frac{s_{i+1,L}}{s_{i,L}} \xrightarrow{L \to \infty} 0,$$

(iii) *For every $1 \leq i \leq r$, if $C^{i,i}$ denotes the top-left $i \times i$ sub-matrix of $C$, then $C^{r,r}$ is positive definite.*

*For each integer $L$, write the SVD of $X_L := S_L C S_L$ as*

$$X_L = U_L \Sigma_L U_L^\top, \quad \Sigma_L = \mathrm{diag}(\sigma_{1,L}, \ldots, \sigma_{n,L}), \quad |\sigma_j| \geq |\sigma_{j+1}| \quad \forall 1 \leq j \leq n$$

*Denote by $e_1, \ldots, e_n$ the standard basis of $\mathbb{R}^n$ and by $u_{1,L}, \ldots, u_{n,L}$ the columns of $U_L$ (which coincide with the singular vectors of $X_L$). We can assume $(e_i^\top u_{i,L}) \geq 0$ for every $1 \leq i \leq n$.*

*Then for all $1 \leq i \leq r$:*

(a) *The singular vector $u_i$ aligns with $e_i$, i.e. $u_i \xrightarrow{L \to \infty} \pm e_i$,*

(b) *With the convention that $C^{0,0} = 1$, then*

$$\frac{\sigma_{i,L}}{s_{i,L}^2} \xrightarrow{L \to \infty} \frac{\det C^{i,i}}{\det C^{i-1,i-1}} =: \ell_i$$

(c) *For all $1 \leq j \leq r$, $j \neq i$,*

$$(e_j^\top u_{i,L}) = \begin{cases} (T_C^{-1})_{i,j} \frac{s_{j,L}}{s_{i,L}} + o\left(\frac{s_{j,L}}{s_{i,L}}\right) & \text{if } j < i, \\ (T_C)_{j,i} \frac{s_{i,L}}{s_{j,L}} + o\left(\frac{s_{i,L}}{s_{j,L}}\right) & \text{if } i < j. \end{cases}$$

*where $T_C$ is the $r \times r$ lower-triangular matrix with all-1 diagonal such that*

$$C^{r,r} = T_C \Sigma_C T_C^\top$$

*is the Cholesky decomposition of $C^{r,r}$ (in which case $\Sigma_C = \mathrm{diag}(\ell_1, \ldots, \ell_r)$).*

*Hence the top-left $r \times r$ sub-matrix $U_L^{r,r}$ converges to the identity matrix, with off-diagonal coefficients decaying faster the further away they are from the diagonal.*

*Remark* D.2. With notations of Theorem D.1, statements $(b)$ and $(c)$ are equivalent to

$$(S_L^{r,r})^{-1} \Sigma_L (S_L^{r,r})^{-1} \xrightarrow{L \to \infty} \Sigma_C$$
$$(S_L^{r,r})^{-1} U_L^{r,r} S_L^{r,r} \xrightarrow{L \to \infty} T_C,$$
$$S_L^{r,r} U_L^{r,r} (S_L^{r,r})^{-1} \xrightarrow{L \to \infty} (T_C^{-1})^\top.$$

*Remark* D.3. Statement *(c)* of Theorem D.1 can be made slightly stronger. Even for $1 \leq i \leq r < j$, there is a constant $K_{j,i}$ such that

$$(e_j^\top u_{i,L}) = K_{j,i} \frac{s_{i,L}}{s_{j,L}} + o\left(\frac{s_{i,L}}{s_{j,L}}\right)$$

but this time the connection between $C$ and $K_{j,i}$ is less clear.

### D.2. Proof of Theorem D.1

We begin by recalling a standard statement from matrix perturbation theory, the proof of which we omit.

**Lemma D.4.** *Let $(M_L)$ be a sequence of symmetric matrices. Let $\sigma_L$ be the largest (in absolute value) eigenvalue of $M_L$, and $u_L$ be a corresponding normalized eigenvector. Assume that*

$$M_L \xrightarrow{L \to \infty} e_1 e_1^\top.$$

*Then $\sigma_L \xrightarrow{L \to \infty} 1$, $u_L \xrightarrow{L \to \infty} \pm e_1$, and all other eigenvalues of $M_L$ converge to zero.*

From this we deduce by induction:

**Lemma D.5.** *Statements (a) and (b) of Theorem D.1 hold. Moreover, $\frac{\sigma_{r+1,L}}{\sigma_{r,L}} \xrightarrow{L \to \infty} 0$.*

*Proof.* For the base case, notice that the matrix $\frac{1}{s_{1,L}^2} X_L$ converges to $e_1 e_1^\top$. By Lemma D.4

$$u_1 \xrightarrow{L \to \infty} \pm e_1, \quad \frac{\sigma_{1,L}}{s_{1,L}^2} \to c_{1,1} = \det C^{1,1} = \ell_1.$$

Assume the statements have been established up to $1 \leq i - 1 < r$. Consider now the matrix

$$\wedge^i(X_L) = (\wedge^i S_L)(\wedge^i C)(\wedge^i S_L)$$

with top-left coefficient given by $\eta_L = \det C^{i,i} \prod_{k=1}^i s_{k,L}^2$. The spectral separation assumption implies

$$\frac{1}{\eta_L}(\wedge^i X_L) \to (e_1 \wedge \cdots \wedge e_i)(e_1 \wedge \cdots \wedge e_i)^\top$$

which shows that by Lemma D.4 again

$$\frac{\prod_{k=1}^i \sigma_{k,L}}{\prod_{k=1}^i s_{k,L}^2} = \prod_{k=1}^i \frac{\sigma_{k,L}}{s_{k,L}^2} \to \det C^{i,i}, \quad u_{1,L} \wedge \cdots \wedge u_{i,L} \to \pm e_1 \wedge \cdots \wedge e_i.$$

By induction the left-hand side yields $\sigma_{i,L}/s_{i,L}^2 \to \ell_i$. Moreover, according to paragraph B.2.3 the subspace $\mathrm{Span}(u_{1,L}, \ldots, u_{i,L})$ aligns with $\mathrm{Span}(e_1, \ldots, e_i)$. By induction, this was already the case for $i - 1$. As $u_{i,L}$ is orthogonal to $\mathrm{Span}(u_{1,L}, \ldots, u_{i-1,L})$, it aligns with the orthogonal complement of $\mathrm{Span}(e_1, \ldots, e_{i-1})$ in $\mathrm{Span}(e_1, \ldots, e_i)$, hence $u_{i,L} \to \pm e_i$.

For the remaining claim, consider the case $i = r$. The second highest eigenvalue of $\frac{1}{\eta_L}(\wedge^r X_L)$ is precisely $\sigma_{r+1,L}/\sigma_{r,L}$, which converges to zero according to Lemma D.4. $\square$

Lemma D.5 establishes in particular that

$$(S_L^{r,r})^{-1} \Sigma_L (S_L^{r,r})^{-1} \xrightarrow{L \to \infty} \Sigma_C.$$

We may now prove the remaining statement.

**Lemma D.6.** *Statement (c) of Theorem D.1 holds.*

*Proof.* Write $C^{r,r} = T_C \Sigma_C T_C^\top$ for the Cholesky decomposition as in the statement. By definition of $X_L$, we have

$$C = S_L^{-1} X_L S_L^{-1} = S_L^{-1} U_L \Sigma_L U_L^\top S_L^{-1} = (S_L^{-1} U_L S_L)(S_L^{-1} \Sigma_L S_L^{-1})(S_L U_L^\top S_L^{-1}). \tag{2}$$

Let $P_L = S_L^{-1} U_L S_L$, which we write in blocks as

$$P_L = \begin{pmatrix} P_L^{r,r} & \varepsilon_L \\ * & * \end{pmatrix}.$$

The spectral separation assumption implies that the $r \times (n-r)$ matrix $\varepsilon_L$ converges to zero. Decomposing the $r \times r$ matrix $P_L^{r,r}$ as

$$P_L^{r,r} = Q_L + R_L$$

where $Q_L$ is lower-triangular with all-1 diagonal and $R_L$ is upper triangular with all-0 diagonal, spectral separation shows once more that $R_L$ converges to 0. A computation performed in Lemma D.7 below yields

$$C^{r,r} = Q_L \Sigma_C Q_L^\top + o(1).$$

In particular, $Q_L \Sigma_C Q_L^\top$ is the Cholesky decomposition for the positive definite matrix $C^{r,r} + o(1)$, which converges to the positive definite matrix $C^{r,r}$. By continuity and uniqueness of the Cholesky decomposition for positive definite matrices, we find

$$Q_L \xrightarrow{L \to \infty} T_C$$

and hence $P_L^{r,r}$ converges to $T_C$. Thus

$$(S_L^{r,r})^{-1} U_L^{r,r} S_L^{r,r} \xrightarrow{L \to \infty} T_C$$

and the remaining convergence is obtained by taking transpose and inverse. $\qquad\square$

**Lemma D.7.** *With notations of the proof of Lemma D.6,*

$$C^{r,r} = Q_L \Sigma_C Q_L^\top + o(1).$$

*Proof.* Define the diagonal blocks

$$S_L^{-1} \Sigma_L S_L^{-1} = \begin{pmatrix} D_{1,L} & 0 \\ 0 & D_{2,L} \end{pmatrix}.$$

From Lemma D.5 we have $D_{1,L} = \Sigma_C + o(1)$. Plugging into 2 yields

$$C^{r,r} = P_L^{r,r} D_{1,L} P_L^{r,r\top} + \varepsilon_L D_{2,L} \varepsilon_L^\top = Q_L \Sigma_C Q_L^\top + \varepsilon_L D_{2,L} \varepsilon_L^\top + o(1).$$

Thus it suffices to show that $\varepsilon_L D_{2,L} \varepsilon_L^\top$ converges to zero. This is an $r \times r$-matrix with coefficient in position $(i,j)$ given by

$$(\varepsilon_L D_{2,L} \varepsilon_L^\top)_{i,j} = \sum_{k=1}^{n-r} \frac{\sigma_{r+k}}{s_{i,L} s_{j,L}} (e_i^\top u_{r+k,L})(e_j^\top u_{r+k,L}).$$

Notice that the terms $(e_i^\top u_{r+k,L})(e_j^\top u_{r+k,L})$ are bounded by 1, and that

$$\frac{|\sigma_{r+k}|}{s_{i,L} s_{j,L}} \leq \frac{|\sigma_{r+1}|}{\sigma_{r,L}} \cdot \frac{\sigma_{r,L}}{s_{r,L}^2} \xrightarrow{L \to \infty} 0$$

according to Lemma D.5. Hence the $(i,j)$-coefficient of $\varepsilon_L D_{2,L} \varepsilon_L^\top$ converges to zero. $\qquad\square$

# E. Proofs of jacobian singular value dynamics

## E.1. Preliminaries

**Lemma E.1.** *Consider an FGLN $J(t) = W_L(t)D_{L-1}\cdots D_1 W_1(t)$ trained by gradient flow on the weights $(W_\ell(t))_{\ell=1}^L$ with loss $\mathcal{L}(J)$. Define $M_\ell(t) := D_\ell W_\ell(t)$ for $\ell = 1,\ldots,L$. Then:*

$$\dot{J} = -\sum_{\ell=1}^L M_{\ell+1:L} D_\ell M_{\ell+1:L}^\top \nabla_J \mathcal{L}(J) M_{1:\ell-1}^\top M_{1:\ell-1}$$

*Proof.*

$$\dot{J} = \sum_{\ell=1}^L M_{\ell+1:L} \dot{M}_\ell M_{1:\ell-1}$$

And for all $\ell$, $\dot{M}_\ell = D_\ell \dot{W}_\ell = -D_\ell \nabla_{W_\ell} \mathcal{L}$ And using lemma A.1 we get the result. $\qquad\square$

The derivative of singular values are related to the derivative of the Jacobian as follows.

**Lemma E.2.** *Let $J$ a matrix depending smoothly on time. Let $J = USV^\top$ be its singular value decomposition. Then:*

$$u_k^\top \dot{J} v_k = \dot{s}_k$$

*Proof.*

$$\dot{J} = \dot{U}SV^\top + U\dot{S}V^\top + US\dot{V}^\top$$

Hence

$$u_k^\top \dot{J} v_k = u_k^\top \dot{u}_k s_k + \dot{s}_k + s_k \dot{v}_k^\top v_k$$

However $u_k^\top \dot{u}_k = \frac{1}{2}\frac{d}{dt}\|u_k\|_2^2 = 0$ and similarly $\dot{v}_k^\top v_k = 0$ thus the result. $\qquad\square$

## E.2. Proof of Proposition 7.1

*Proof of Proposition 7.1.* We work at a fixed time $t$. Note that for all $\ell$ we have $J = A_\ell W_\ell B_\ell$. Applying Lemmas E.1 and E.2 yields

$$\dot{s}_k = -\left(\sum_{\ell=1}^L U^\top A_\ell A_\ell^\top \nabla_J \mathcal{L} B_\ell^\top B_\ell V\right)_{k,k}$$

Incorporating our notations and assumptions, this becomes

$$\dot{s}_k = -\sum_{\ell=1}^L (U^\top U_{A_\ell} S_{A_\ell}^2 U_{A_\ell}^\top U \left(U^\top \nabla_J \mathcal{L} V\right) V^\top V_{B_\ell} S_{B_\ell}^2 V_{B_\ell}^\top V)_{k,k}.$$

Since we only care about the first order in $\varepsilon$ we may assume $U^\top U_{A_\ell} = V^\top V_{B_\ell} = I$ by assumption *(ii)*. In this case the equation becomes

$$\dot{s}_k \overset{\varepsilon\to 0}{\sim} -\sum_{\ell=1}^L s_{k,A_\ell}^2 s_{k,B_\ell}^2 \left(U^\top \nabla_J \mathcal{L} V\right)_{k,k}$$

Assumption *(i)* ensures $s_{A_\ell,k} s_{B_\ell,k} = e^{\delta_k - \gamma_k} s_k = e^{(1+\frac{1}{L})\delta_k} s_k^{1-\frac{1}{L}}$. Thus:

$$\dot{s}_k \overset{\varepsilon\to 0}{\sim} -e^{(2+\frac{2}{L})\delta_k} s_k^{2-\frac{2}{L}} \sum_{\ell=1}^L \left(U^\top \nabla_J \mathcal{L} V\right)_{k,k} = -e^{(2+\frac{2}{L})\delta_k} L s_k^{2-\frac{2}{L}} \left(U^\top \nabla_J \mathcal{L} V\right)_{k,k}$$

The conclusion follows from the observation that

$$\left(U^\top \nabla_J \mathcal{L} V\right)_{k,k} = \langle \nabla_J \mathcal{L}, u_k v_k^\top \rangle.$$

$\qquad\square$

### E.3. Proof of Proposition 7.2

*Proof of Proposition 7.2.* Since $J = A_\ell M_\ell B_\ell$ for any $\ell$, the derivative of the Jacobian reads

$$\dot{J} = \sum_{\ell=1}^{L} A_\ell \dot{M}_\ell B_\ell.$$

Let $1 \leq k \leq n$. By Lemma E.2 we can access $\dot{s}_k$ via

$$\dot{s}_k = (U^\top \dot{J} V)_{k,k} = \sum_{\ell=1}^{L} \left( U^\top U_{A_\ell} S_{A_\ell} V_{A_\ell}^\top \dot{M}_\ell U_{B_\ell} S_{B_\ell} V_{B_\ell}^\top V \right)_{k,k}.$$

Since we only care about the first order in $\varepsilon$, by assumption *(ii)* we may assume

$$S_{A_\ell}^{-1} U^\top U_{A_\ell} S_{A_\ell} = T_\ell^-, \qquad S_{B_\ell} V_{B_\ell} V^\top S_{B_\ell}^{-1} = T_\ell^+.$$

Then

$$\dot{s}_k \overset{\varepsilon \to 0}{\sim} \sum_{\ell=1}^{L} (S_{A_\ell} T_\ell^- V_{A_\ell}^\top \dot{M}_\ell U_{B_\ell} T_\ell^+ S_{B_\ell})_{k,k}$$

$$= \sum_{\ell=1}^{L} s_{A_\ell,k} s_{B_\ell,k} (T_\ell^- V_{A_\ell}^\top \dot{M}_\ell U_{B_\ell} T_\ell^+)_{k,k}$$

Assumption *(i)* ensures $s_{A_\ell,k} s_{B_\ell,k} = e^{\delta_k - \gamma_k} s_k = e^{(1+\frac{1}{L})\delta_k} s_k^{1-\frac{1}{L}}$ which is independent of $\ell$ and can be factored out of the sum. This concludes the proof. $\square$

## F. Diagonal correlation coefficient

The *diagonal correlation coefficient*, denoted $\rho$, is a measure of how "diagonal" a square matrix is. It ranges from $-1$ to $1$, attains $1$ (resp. $-1$) precisely for diagonally-supported (resp. antidiagonally-supported) matrices.

Let $A$ be a nonzero $n \times n$ matrix with real entries; the computation of $\rho(A)$ is as follows. Up to taking absolute value we may assume that its coefficients are non-negative. We may now interpret $A_{ij}$ as the weight (or frequency) associated with the joint occurrence of indices $(i, j)$ in a random sampling of an $n \times n$ checkerboard. Define the coordinate random variables $(X, Y)$ with respect to this distribution:

$$\mathbb{P}(X = i, Y = j) = \frac{A_{ij}}{\sum_{k,\ell} A_{k\ell}}.$$

Then $\rho(A)$ is defined as the Pearson correlation coefficient between variables $X$ and $Y$:

$$\rho(A) = \frac{\mathrm{Cov}(X, Y)}{\sigma_X \sigma_Y}.$$

To express it purely in terms of the matrix $A$, we introduce

$$\mathbf{1} = (1, 1, \ldots, 1)^\top, \qquad \mathbf{r_1} = (1, 2, \ldots, n)^\top, \qquad \mathbf{r_2} = (1^2, 2^2, \ldots, n^2)^\top.$$

Let $m = \mathbf{1}^\top A \mathbf{1}$ denote the total mass of $A$. With a little work, one can show that

$$\rho(A) = \frac{m \, (\mathbf{r_1}^\top A \mathbf{r_1}) - (\mathbf{r_1}^\top A \mathbf{1})(\mathbf{1}^\top A \mathbf{r_1})}{\sqrt{(m \, (\mathbf{r_2}^\top A \mathbf{1}) - (\mathbf{r_1}^\top A \mathbf{1})^2) \, (m \, (\mathbf{1}^\top A \mathbf{r_2}) - (\mathbf{1}^\top A \mathbf{r_1})^2)}}.$$

# G. Additional Experiments

### G.1. FGLN Singular value dynamics

We train a FGLN of depth $L = 10$, width $n = 64$ and Bernoulli parameter $p = 0.5$ on a random linear synthetic dataset. This dataset generates 1000 random input gaussian vectors and labels them through a fixed random linear map of rank 10. Both the input and output are of size $n$. We used a classic SGD optimizer with the MSE loss.

In this setup we compare the predicted top singular value trajectory versus the experimental one. The scaling factor in front of the prediction is fine-tuned to match the experiment, as it depends on several parameters that have not been considered in our study (such as the learning rate). Denoting $g(t) = \left\langle \nabla_J \mathcal{L}(J(t)), u_1(t) v_1(t)^\top \right\rangle$ where $u_1$ and $v_1$ are the first left and right singular vector of the full jacobian, we consider the iterative process:

$$s(t+1) = s(t) + C s(t)^{2 - \frac{2}{L}} g(t)$$

where $C$ is the fine tuned scaling factor. We observe in figure 8 that the predicted process matches the shape of the actual singular dynamics.

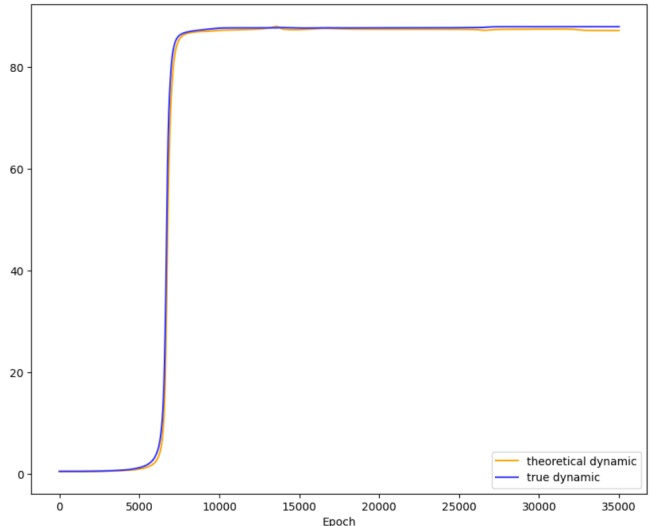

*Figure 8.* Theoretical versus experimental singular value dynamics for $s_1$.

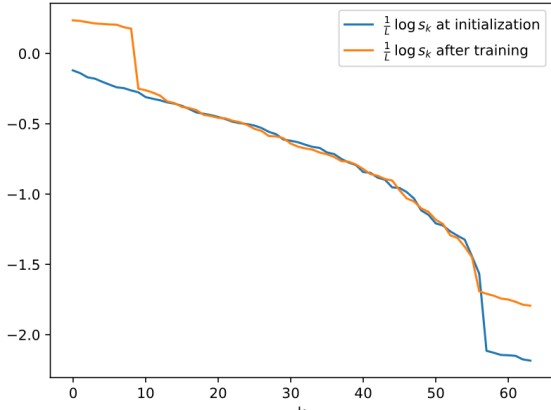

*Figure 9.* We compare the trained singular spectrum with respect to the one at its initialization. We observe that only the top singular values were strongly impacted by the training.

## G.2. Depth scaling and Alignment

Here we consider a square FGLN at initialization with depth $L = 20$ and varying Bernoulli parameter $p$ and width $n$.

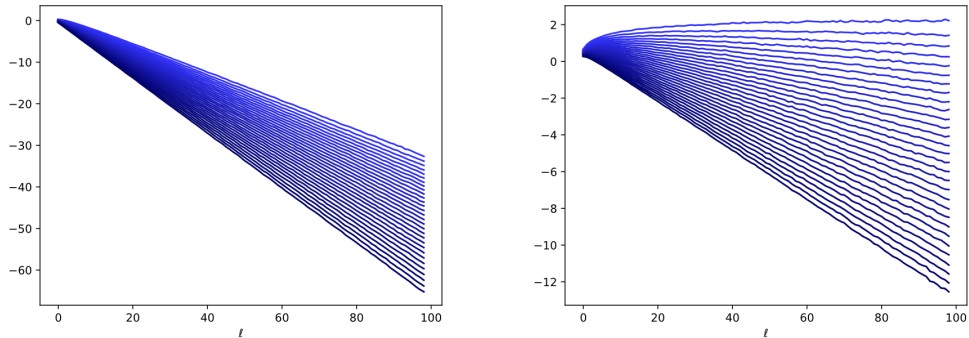

*Figure 10.* Top 30 log singular values in an initialized FGLN with parameters $n = 128$, $p = 0.5$ (left) and $p = 1$ (right)

We investigate the diagonal correlation of the product $U_{J_L}^\top A_\ell A_\ell^\top U_{J_L}$, which is found in the proof of Proposition 7.1. The clustering of $U_{J_L}^\top U_{A_\ell}$ around the diagonal when $\ell \ll L$ induces a similar effect for $U_{J_L}^\top A_\ell A_\ell^\top U_{J_L}$ by virtue of the SVD. However we see in Figure 11 that diagonal clustering for this larger product also occurs to a lesser extent for $\ell$ close to $L$.

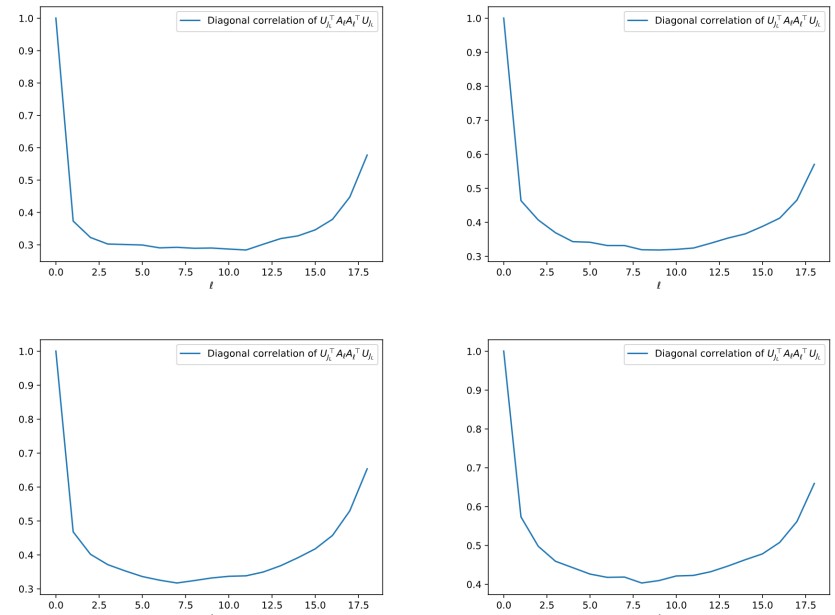

*Figure 11.* Diagonal correlations for the top-left $10 \times 10$ sub-matrix of $U_{J_L}^\top A_\ell A_\ell^\top U_{J_L}$ for parameter $p = 0.5$ (first column), $p = 1$ (second column), $n = 64$ (first row) and $n = 128$ (second row)

This U-shaped profile is explained by two different mechanisms: the diagonal clustering of $U_{J_L}^\top U_{A_\ell}$ when $\ell \ll L$ (Figure 12), as well as a diagonal clustering of $A_\ell A_\ell^\top$ for $\ell \simeq L$ (Figure 13).

We interpret this second phenomenon as follows. We say that a matrix $M$ has *isotropic spectrum* if its singular values $s_k$ are approximately equal. In the extreme case where $s_k = s$ for all $s$, we find $MM^\top = s^2 I$ to be perfectly diagonal; continuously altering the spectrum away from isotropy decreases the diagonal correlation of $MM^\top$. In our setting, as $\ell$ decreases from $L$ to lower values the spectrum of $A_\ell A_\ell^\top$ exhibits stronger spectral separation, which explains Figure 13.

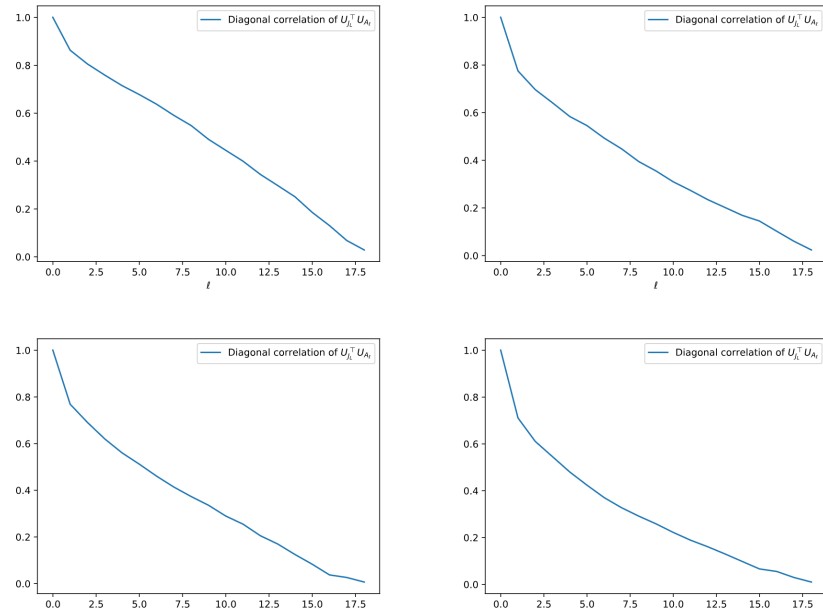

*Figure 12.* Diagonal correlations for the top-left $10 \times 10$ sub-matrix of $U_{J_L}^\top U_{A_\ell}$ for parameter $p = 0.5$ (first column), $p = 1$ (second column), $n = 64$(first row) and $n = 128$(second row)

## G.3. General dynamics

We provide additional experiments to analyse when the expression from Proposition 7.2 effectively induces spectral separation.

We consider a deep linear network $J_L = W_L \cdots W_1$ of $d \times d$ square matrices with prescribed weight dynamics $\dot{W}_\ell$ not necessarily induced from gradient descent. Here, $L = 10$ and $d = 64$. Weights at $t = 0$ are scaled random gaussian matrices, i.e. with $\sigma = 1/\sqrt{d}$. We perform $T = 1000$ weight updates given by $W_\ell(t+1) = W_\ell(t) + \eta \dot{W}_\ell(t)$ where $\eta = 0.01$.

From Proposition 7.2 the dynamics of the $k$-th singular value $s_k$ of $J_L$ is approximately

$$\dot{s}_k \approx e^{(1+\frac{1}{L})\delta_k} s_k^{1-\frac{1}{L}} \sum_{\ell=1}^{L} (T_\ell^- V_{A_\ell}^\top \dot{W}_\ell U_{B_\ell} T_\ell^+)_{k,k} \tag{3}$$

Scenario 1 in Figure 14 prescribes $\dot{W}_\ell = R_\ell$ as a random gaussian matrix: no further separation is observed as expected.

Scenario 2 in Figure 15 prescribes $\dot{W}_\ell := V_{A_\ell} U_{B_\ell}^\top$ aligned with the left and right context matrices $A_\ell$ and $B_\ell$ for all $\ell$, making the inner terms in the summand of Equation 3 diagonal. Spectral separation is observed.

Scenario 3 in Figure 16 prescribes $\dot{W}_\ell := R_\ell + V_{A_\ell} U_{B_\ell}^\top$, notations as above. The result is a combination of both previous scenarios, notably a weaker spectral separation is observed. This suggests that scenario 2 is robust to noise: having weight dynamics partially aligned with context matrices should be enough to induce spectral separation.

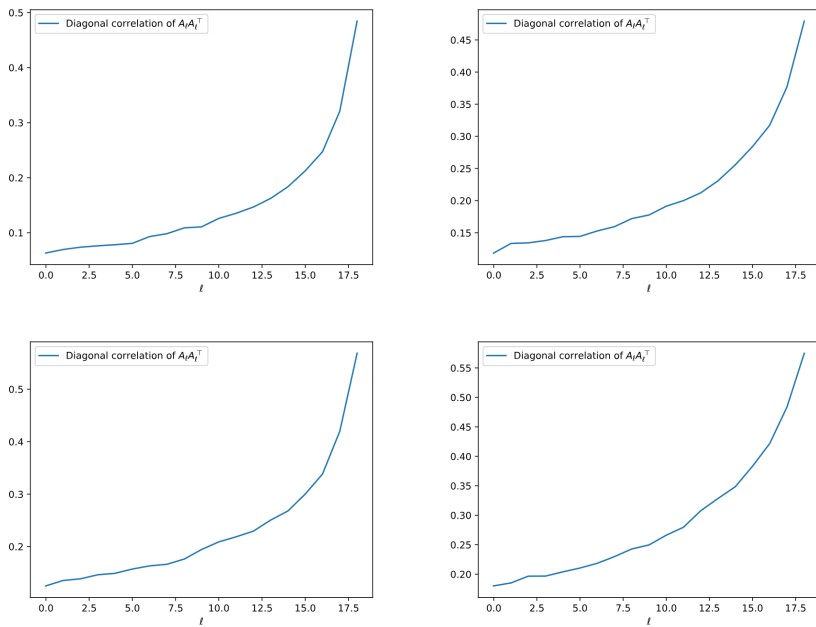

*Figure 13.* Diagonal correlations for the top-left $10 \times 10$ sub-matrix of $A_\ell A_\ell^\top$ for parameter $p = 0.5$ (first column), $p = 1$ (second column), $n = 64$(first row) and $n = 128$(second row)

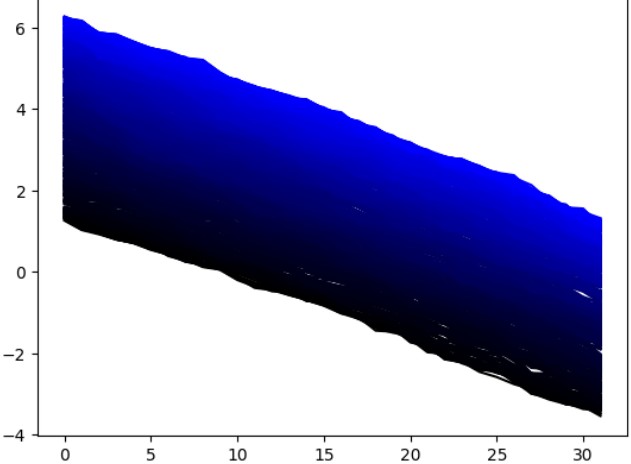

*Figure 14.* First 32 ordered log singular values of $J_L$ for each train step, from black ($t = 0$) to blue ($t = T$), in the first scenario

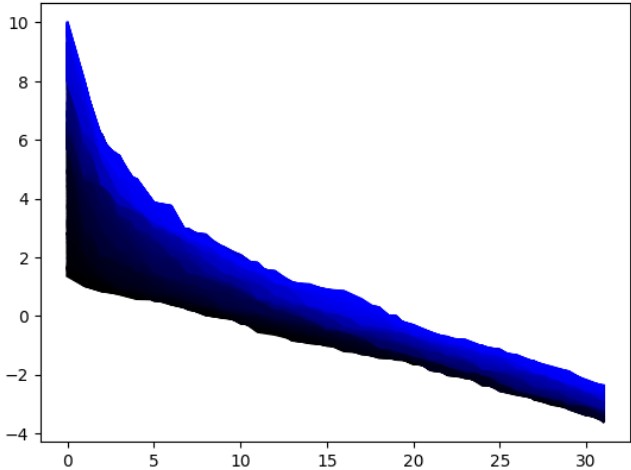

*Figure 15.* First 32 ordered log singular values of $J_L$ for each train step, from black ($t = 0$) to blue ($t = T$), in the second scenario

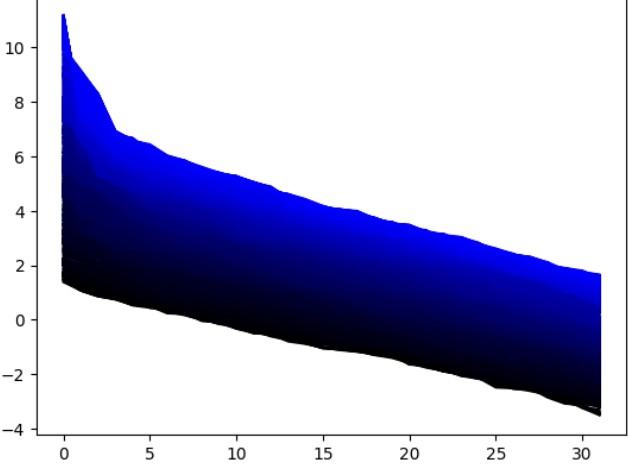

*Figure 16.* First 32 ordered log singular values of $J_L$ for each train step, from black ($t = 0$) to blue ($t = T$), in the third scenario

