# OpenReview forum: "Why Deep Jacobian Spectra Separate: Depth-Induced Scaling and Singular-Vector Alignment"
_ICML.cc/2026/Conference — ICML 2026 spotlight_

### Official Review · Reviewer_k9HF · 2026-03-03

**Soundness:** 4
**Presentation:** 3
**Significance:** 3
**Originality:** 3
**Overall Recommendation:** 5
**Confidence:** 3

**Summary:**

In this work, the authors tackle the problem of singular value dynamics in piecewise linear models. To do so, the authors propose a theoretical analysis based on two main steps: exponential dependence of singular values on the depth of the network and strong spectral separation. In particular, the authors prove the existence of Lyapunov exponents for a structured family of initializations (together with a closed-form expression and finite depth corrections in expectation), and left singular vector alignment in the large depth limit.
These two results together are then used to motivate an approximation of the singular value dynamics in which each one is effectively decoupled from the other, as in the classical deep-linear analysis with balancing assumption.

**Compliance With Llm Reviewing Policy:**

Affirmed.

**Final Justification:**

The authors were able to clarify all my doubts during the rebuttal; my initial recommendation of acceptance remains unchanged.

**Key Questions For Authors:**

The only point I am personally skeptical on is the significance of the result presented in Remark 7.2. While I understand the strong limitation of the result in Proposition 7.1, I am not sure how the result proposed in Remark 7.2 helps, as effectively $\dot M_l$ can be anything. The authors claim the result becomes interesting when looking at dominating terms in the expansion of $\dot M$, but I believe an example here could be of help to understand what happens in some controlled cases.

**Limitations:**

yes

**Strengths And Weaknesses:**

**Soundness**. The manuscript is theoretically sound, and the technical presentation in the appendix is dense but well-written and clear. The authors transparently discuss all hypotheses together with their limitations, providing a very honest overview of what can be improved in the current version of the results.

**Presentation**. The work is overall well written, the presentation is clear, and there is a full discussion of relevant literature.

**Significance**. The problem tackled by the authors is of high interest to the community, as it advances our current theoretical understanding of singular value dynamics in a regime larger than deep learning and detaching from the balanced initialization hypothesis.

**Originality**. The view proposed in the manuscript is novel, clearly distinguished from prior literature on the topic.

---

> ### Author Rebuttal · Authors · 2026-03-30
>
> We thank the reviewer for the careful reading and constructive feedback.
>
> > "The significance of Remark 7.2 is unclear — since [...] can be anything, how does the result help? An example in some controlled case would clarify what happens with dominating terms in the expansion."
>
> This is a fair point. The goal of Remark 7.2 is to suggest that the implicit biais may be a byproduct of working with factorized architectures: already there is a $s^{1-1/L}$ factor appearing in front of the expression for singular dynamics. In the specific scenario of an FGLN with MSE, we can understand the dynamics of $\dot{M_{\ell}}$ explicitly and replace it with an expression in which a second $s_k^{1-1/L}$ factor appears. This suggests that the initial $s^{1-1/L}$ factor (the one appearing in Remark 7.2) should not be compensated by the dynamics of $\dot{M_{\ell}}$ in general. While this is not a rigorous argument (perhaps there exist specific architectures and loss landscapes where the initial $s^{1-1/L}$ factor gets damped; this could be a topic for future work), we hope that the general aspect of the singular value dynamics displayed in Remark 7.2 illustrates the interplay between factorized architecture and implicit bias. We did not try to investigate further examples in which $\dot{M_{\ell}}$ is tractable; this could be material for future work. We will extend Remark 7.2 to justify its significance along these lines.

---

> > ### Author Rebuttal · Reviewer_k9HF · 2026-04-01
> >
> > I thank the authors for the honest answer. I believe it could be helpful to include a small remark about this fact in the revised version of the manuscript. My doubts have been resolved, therefore I will keep my original full acceptance score.

---

> > > ### Author Response · Authors · 2026-04-06
> > >
> > > We thank the reviewer for the careful reading of the positive feedback.  We are pleased that our clarifications addressed your concerns.  We also appreciate the suggestion and will include a remark on this point in the revised manuscript.

---

### Official Review · Reviewer_FXhS · 2026-03-10

**Soundness:** 3
**Presentation:** 4
**Significance:** 3
**Originality:** 4
**Overall Recommendation:** 5
**Confidence:** 4

**Summary:**

This work recovers deep-linear-like singular-value evolution, under random initialization and with a gated-product structure for layer-wise Jacobians, substituting the balancing hypothesis, which necessarily ties each layer to the next, for depth-induced scaling. In the authors’ setting, the Jacobian reduces to a product of weight matrices interleaved with diagonal gates, and, more specifically for this work, fixed gates and masked linear networks (i.e., 0/1 diagonal entries). The theoretical findings establish and empirically support spectral separation for such Jacobians. Strong spectral separation is then sufficient to force alignment of dominant singular vectors. This yields an approximation regime where singular-value evolution becomes effectively decoupled (per $k$-th singular value). Taken together, this supports emergent low-rank Jacobian structure as a driver of implicit bias in deep networks.

**Compliance With Llm Reviewing Policy:**

Affirmed.

**Key Questions For Authors:**

* Fig 6: What is the intuition for getting fuzziness at bottom left corner for $L=20$ and $\ell=L/2$  as opposed to $L=2,10$?
* Though the paper results are produced for FGLNs at initialization, training assumes gates are kept fixed. Is this done to maintain control over rank? Or for some other purpose?
* How does the plausibility of the two assumptions  (depth scaling, spectral separation) compares to that of balancing?
* I understand the need to keep gate rank bounded from below away from 0. What NN architectures would asymptotics conditioned as such inform?
* Assuming depth scaling, for spectral alignment we need consecutive Lyapunpov exponents gap to be bounded from below. What is the intuition for that to happen in experimental settings where spectral separation has been observed?

**Limitations:**

The authors adequately discuss the limitations of their work.

* Authors discuss the limits of single-mode fixed-gates in Remark 7.2 and Sec. 9.
* Authors acknowledge that the “spectral separation hypothesis only holds in the unrealistic regime” of $s_{k+1}/s_k \to 0$.
* Authors are explicit that Theorems 5.3 and 6.1 are not built into a training-time statement; rather, they motivate the approximation regime used in Proposition 7.1.

**Strengths And Weaknesses:**

Soundness:
=========

Though I haven't read the supplemental material in depth, I find the logical flow of the manuscript convincing. One related questions:

* The manuscript’s Theorem 5.3 is based on a result by Bougerol et al. (2012). The authors observed that the assumption in Bougerol et al. that the matrices $Y_i$ need be invertible can be relaxed to $J_L$ almost surely nonzero. The authors state that reproducing Bougerol et al.’s argument in the appendix would be excessively long. Is it possible to provide high-level justification, stating the critical point in the proof where invertibility is employed and why/how it could be relaxed?

Presentation:
==========

Presentation is superb. The submission is clearly written and well structured, and its overall narrative is easy to follow. The work properly positions itself within the study of implicit bias in deep NNs via the study of deep Jacobian spectra via the study of random matrix factorization.

Mild possible typos/adjustments:
* In Def 4.6 $\epsilon \ge 0$?
* Matrices are square in Def 4.7?
* Def 5.1 add space after i.i.d.
* The 5.3 specify expectation over what?
* The use of log and ln (eg, Remarks 5.4-5): what is the base for the log?
* First sentence of Sec. 7: “separation, induced” —> “separation-induced”?
* Proposition 7.1: (i) is assuming $\epsilon=0$-depth scaling, no?
* Remark 7.2: What is $(\cdot)_{k,k}$?
* In Def 4.6, matrices are assumed square? And in Prop 4.5?

Significance:
==========

This paper addresses the important question of implicit bias in deep networks. Its scope is more specialized, given that it focuses on models of Jacobian in piecewise-linear networks. Its two key assumptions -- depth scaling and spectral separation -- are established at initialization, while the authors acknowledge that, dynamically, they are not expected to hold uniformly across all layers.

## Pros
* The manuscript studies ordered singular values at finite depth in gated product models and formalizes sufficient conditions for spectral separation that do not rely on infinite-width limits or isometry assumptions.
* The manuscript focuses on finite depth NNs, beyond asymptotic or equilibrium characterizations.
* Neural Jacobians often take the form of products of matrices with diagonal gating operators in between that may be singular, placing them outside standard invertible-product theory (e.g., the multiplicative ergodic theorem). This work extends Lyapunov-type reasoning to this setting.

## Cons
* Results and experiments established at initialization (apart from Fig. 1).
* Only Fig 1. seems to put the spectral separation assumption to the test during training.
* If I understand correctly, results apply to single-mode fixed-gate linear networks (FGLNs), while an MLP without biases is a multi-modes FGLN.

Originality:
========

The authors make a convincing case for how their contributions extend beyond prior art.

*  They prove depth scaling, including finite-depth corrections, for masked products at random initialization not assuming balancing (Theorem 5.3). It goes beyond prior art, which establishes Lyapunov exponents for products of random invertible i.i.d. matrices. In this case $D_lW_l$ are allowed to be singular. Since the diagonals are samples from Bernoulli distributions we nevertheless remain within the Masked Linear Network model, while the Gaussian i.i.d. assumption for the weights is standard.
* Complementing classical matrix perturbation theory, they show that strong spectral separation asymptotically (in depth) forces dominant singular-vector alignment in matrix products (Theorem 6.1).
* Assuming depth scaling and spectral separation-induced alignment, they obtain a deep-linear-like singular-value evolution for fixed-gates networks (Proposition 7.1)

---

> ### Author Rebuttal · Authors · 2026-03-30
>
> We thank the reviewer for the careful reading and constructive feedback.
>
> > 1. Theorem 5.3 - Bougerol et al
>
> Thm. I.4.1 in Bougerol et al. (2012) relies on Furstenberg and Kersten (1960) (this had been previously overlooked on our part), whose result also applies here for i.i.d. matrix sequences under mild moments. We will clarify.
>
> The invertibility assumption simplifies the proof. Bougerol et al. define $Y \cdot M = YM/|YM|$ for $Y \in GL_d$ and $M \in B$. For arbitrary $Y$, $YM$ may vanish, so the action is ill defined. Furstenberg and Kersten address this by adjoining $0$ to $B$ and setting $Y \cdot M=0$ when $YM=0$. Under mild assumptions on the law of $Y$, the set ${(Y,M): YM=0}$ has measure zero, so the proof integrals are unchanged. The remaining points, e.g. cocycle definitions and a.e. inequalities, are handled similarly.
>
> > 2. Fuzziness in Fig. 6
>
> The bottom-left fuzziness is due to rank deficiency. When $p<1$, gate matrices have rank about $np<n$, so $J_L$ has at most $np$ nonzero singular values. The remaining singular vectors correspond to near-zero singular values and are numerically unstable, so  SVD may return arbitrary orthonormal directions, producing noisy block. This is expected: relevant part is the top $r \lesssim np$ singular vectors, where Thm. 6.1 predicts clean diagonal alignment.
>
> The figure shows top-left $64 \times 64$ block of the full $128 \times 128$ matrix $U_{J_L}^\top U_{A_\ell}$. For shallow products, rank is close to $np \approx 64$, so fuzzy region lies outside the displayed block. As depth increases, rank decreases and fuzzy region enters from the bottom-left corner. It stays confined there and cannot affect the diagonal block above because null-space singular vectors are orthogonal to those for nonzero singular values.
>
> > 3. Fixed gates vs ReLU MLPs
>
> We use fixed gates to isolate the spectral mechanism. At init., Thm. 5.3 applies to any piecewise-linear network, including MLPs. During training, our analysis is for an infinitesimal gradient-flow step, over which gates remain fixed. This is the regime for testing the singular-value dynamics of Prop. 7.1.
>
> In a true ReLU MLP, gate patterns change with the weights, causing discrete changes in Jacobian spectrum when neurons switch activation state. We do not expect this to alter within-mode dynamics, but spectral continuity across transitions is left open. Real MLPs also involve multiple coexisting gate patterns whose gradient updates are coupled across inputs. This coupling is absent here and makes full multi-mode analysis harder. Extending the theory in that direction, following Def. 4.3, is a next step.
>
> > 4. Balancing vs depth scaling
>
> Balancing is useful theoretically: it is exactly preserved under gradient flow (Prop. 4.5) and yields explicit singular-value dynamics in deep linear nets. But it must hold at init., excluding random initializations; it is not exactly preserved at finite learning rate; and the proof is specific to deep linear architectures, with no clear extension to nonlinear settings.
>
> By contrast, depth scaling and spectral separation arise naturally in factorized architectures at init. (Thm. 5.3, Cor. 5.6), and can be monitored empirically in training across broader architectures. Similar effects observed in CNNs and Transformers by Oymak et al. (2019) and Huang et al. (2025), making them a stronger basis for general analysis. Prop. 7.1, Rem. 7.2, and Thm. 6.1 suggest that spectral separation is self-reinforcing in training. No analogous mechanism is known for balancing.
>
> > 5. Condition involving r
>
> We are unsure what is meant by “asymptotics conditionned as such.” $r$ is the number of tracked top singular values, typically on the order of the output dimension and usually $r \ll n$. Once $r \ll np$ holds, an $(n,p)$-gate has rank at least $r$ w.h.p. With $n=256$ and $r=10$, this is roughly the case for $p>0.1$. In that regime, such a gate is effectively indistinguishable from an $(r,n,p)$-gate, so our result applies. The lower rank bound is mainly a theoretical requirement for rigorous proofs, not a practical limitation.
>
> > 6. Separation of Lyapunov exponents
>
> If the question is how to assess whether the Lyapunov exponents in the depth-scaling profile are sufficiently separated for a given NN (e.g. FGLN), our theory gives a criterion. Random matrix theory shows that products of i.i.d. randomly oriented matrices with anisotropic spectra satisfy depth scaling with distinct Lyapunov exponents, with gaps roughly controlled by spectral anisotropy. Thus, at init., anisotropic spectra imply exponent separation.
>
> For trained nets, this is no longer exact due to induced correlations. For short training, anisotropy of each matrix spectrum remains useful proxy. For more strongly trained linear nets, one expects singular vectors across layers to align, so the I/O Jacobian spectrum is approximated by the product of individual spectra. Hence, anisotropy of the layer spectra suggests distinct Lyapunov exponents.

---

> > ### Author Rebuttal · Reviewer_FXhS · 2026-04-02
> >
> > The authors have addressed all my concerns satisfactorily. I think the results are novel and solid theoretically. My score remains 5.

---

> > > ### Author Response · Authors · 2026-04-06
> > >
> > > We thank the reviewer for the careful reading of the positive feedback.  We are pleased that our clarifications addressed your concerns.

---

### Official Review · Reviewer_g5Zs · 2026-03-11

**Soundness:** 4
**Presentation:** 3
**Significance:** 4
**Originality:** 3
**Overall Recommendation:** 5
**Confidence:** 3

**Summary:**

The paper addresses the question of the implicit bias in deep learning
from the angle of spectral analysis of Jacobian matrices. To this end,
the authors propose a theoretically analyzable model of deep learning
which extends earlier approaches of deep linear networks. The novel
ingredient are gating layers, which represent the presence of the
derivative of the gain funciton. In particular, they study such gating
matrices with Bernoulli random entries on the diagonal.
The main results of the paper is a mathematically rigorous (as far as I
can tell) analysis of the spectral properties of such gated linear
networks.
Specifically, the authors investigate
* depth scaling of singular
  values, where from a cutoff rank r on singular values decline quickly
  (separation)

* explicit expressions for the singular value spectra of randomly initialized
  gated linear networks, where gate variables are Bernoulli

* the effect of alignment of singular value spaces induced by the separation
  of singular values, so that left and right sided singular spaces of the
  full Jacobian align with spaces of left and right-sided sub-products of
  the matrices that compose the entire Jacobian

**Compliance With Llm Reviewing Policy:**

Affirmed.

**Key Questions For Authors:**

Please see the numbered points above.

**Limitations:**

Yes.

**Strengths And Weaknesses:**

Soundness:
As far as I can tell, the results appear sound; I am, however, not an
expert in random matrix theory and correspondingly did not check the
proofs in detail.

Presentation:
The paper is well structured and constructed in a manner that is
most suitable for a readership with a math background. At times I
was wondering if all definitions are indeed needed. For example, the
paper mainly investigates networks where gates are Beroulli, but
initially introduces also gates with real values.

Minor and specific points regarding presentation:

1.1 - For the result of Theorem 5.3 (and also Figure 3),
  it would be good to state that the result for \gamma_i
  with p=1 corresponds to the non-masked case. Also it may
  be helpful for the reader to state that for p=1
  (and in the high p regime r << np) only a single
  term \Psi(n-i+1/2) remains in the sum.

1.2 - I was a bit confused by the definition of M_\ell(t) = D_\ell^T
  W_\ell D_{\ell-1} in proposition 4.5, since we want to write J as a
  product of Ms.  If the idea here to exploit D^2 = D, valid in the
  masking case? This could be stated explicitly to help the
  reader understand the results

1.3 - Which loss function has been used for the numerical experiments?
  It would be helpful to be stated in conjuction with the figures
  (e.g., in the caption)



Significance and soundness:
The paper is highly significant as it provides rigorous insights
into the spectral properties of Jacobian matrices in well-controlled
toy models.
What would strengthen the work considerably would be experiments
using ReLU activation functions (which would nicely correspond
to the masking case investigated here) to show that even if
masking is not fixed, but a function of the preactivations,
some of the results derived here still hold true.

Specific points in this context:

2.1 - It would be helpful to clarify the Remark 4.4,
  stating that a MLP is indeed a multi-mode fixed gate network. Is MLP
  here referring to a true MLP with non-linear activation function f,
  so  f(W (f (W ... f(Wx) ) ) )?
  What I fail to see in this context is why the gates would be fixed in
  this case along the training trajectory, as the derivative f' in
  general changes during training. Likely this is a misunderstanding of
  mine of the notation.


2.2 - I was wondering how important all terms in the sum over the
  binomially distributed value of t are in the result
  for \gamma_i in Remark 5.4.

  Would one not expect that in the high-n limit, the term
  t ~ n p dominates the result?
  This would be quite intuitive, because effectively the 0 of
  the masking erase dimensions so that the "effective dimension"
  is approximately np.
  Also may also be worth noting that the result including the sum
  may be seen sa a superposition of non-masked networks with dimension n=t,
  each appearing with a binomial probability; at least to provide
  some intuition.

Originality:
The paper proposes an interesting and highly extension to the
literature on training dynamics in deep linear networks.
It builds on top of an established body of work.

---

> ### Author Rebuttal · Authors · 2026-03-30
>
> We thank the reviewer for the careful reading and constructive feedback.
>
>
> > "1.1 — For Theorem 5.3 (and Figure 3), it would be good to state that the result for γ_i with p=1 corresponds to the non-masked case, and that for p=1 only a single ψ term remains in the sum."
>
> We fully agree. We will add this statement to the Remarks following Theorem 5.3 and a reference to Newman 1986.
>
>
> > "1.2 — I was confused by the definition of $M_\ell(t) = D_\ell W_\ell D_{\ell-1}$ in Proposition 4.5. Is the idea to exploit $D^2 = D$, valid in the masking case? Could this be stated explicitly?"
>
> This is correct, we will state it more explicitly as it is part of the motivations for this definition. This relies heavily on the idempotent property of $D_{\ell}$ in the masking case.
>
>
> > "1.3 — Which loss function has been used for the numerical experiments?"
>
> We use CrossEntropy for the MNIST setup and MSE for the synthetic dataset plotted in the appendix. This has been overlooked on our part and will be explicitly mentioned alongside the other hyperparameters used in both experiments.
>
>
> > "2.1 — Could you clarify Remark 4.4, stating that an MLP is a multi-mode fixed-gate network? Why would the gates be fixed along the training trajectory when the derivative f′ changes during training?"
>
> We thank the reviewer for this clarification request. Remark 4.4 introduces the multi-mode FGLN as a model for MLPs. In this framework, each mode corresponds to a specific gate pattern — i.e., a specific configuration of active/inactive neurons — and the network is equipped with a mode matching function $\sigma(x, t)$ mapping each input $x$ to the mode it activates at time $t$. In a real ReLU MLP, this mode matching function is not fixed: as the weights $W_\ell(t)$ evolve during training, the pre-activations change, which changes the gate pattern activated by each input $x$, and therefore $\sigma(x, t)$ itself changes throughout training. This is the key distinction between a true MLP and a single-mode FGLN, where the gate pattern is frozen.
> The space of modes can nonetheless be considered independently of the mode matching function: a mode is simply a choice of binary gate pattern $(D_1, \dots, D_{L-1})$, and nothing prevents us from studying how the weights evolve within a fixed mode, regardless of which inputs realize it.
> We note however that Proposition 7.1 is not directly equivalent to such a per-mode analysis of an MLP: it assumes that the entire training dynamic is governed by a single fixed gate pattern, which is a stronger assumption than what holds in practice. The result that genuinely applies to an individual mode of an MLP is Remark 7.2, which works in the more general setting where the weight dynamics $\dot M_{\ell} $ are left completely unspecified, the formula holds for any choice of $\dot{M}_\ell$, and in particular applies to one mode of a multi-mode FGLN independently of its realization by a specific input. A mode transition — triggered by a neuron switching activation state as weights change — can be viewed as a jump from one mode to another, each governed by the same underlying equations. The theoretical study of these transitions, and in particular whether the singular value spectrum changes continuously across them, is an interesting direction that we leave for future work.
> We will clarify those important considerations in both Remark 4.4 and the discussion section.
>
> > "2.2 — How important are all the terms in the sum over the binomially distributed value of t in Remark 5.4? Could the result be interpreted as a superposition of non-masked networks with dimension n=t?"
>
> The formula for $\gamma_i$ is obtained as the expectation over (binomially-distributed) $t$ of the $i$-th Lyapunov exponent in the case where the rank of all gates is set to $t$. The computation for fixed $t$ is performed in Lemma C.3, we then take the expectation in Lemma C.4. It follows that in the high-$n$ limit, the leading terms in the sum are indeed concentrated around $t \sim np$ (the slight shift due to the digamma factor is insignificant). The interpretation as a superposition of exponents for networks with varying shapes is quite interesting and makes a lot of sense: Lemma C.3 boils down to computing the expectation of $||DW||$ (or of an exterior power thereof), with $D$ a fixed gate of rank $t$, and $W$ random $\sigma$-Ginibre, both of size $n$, and it is not too hard to show that this coincides with the norm of a $\sigma$-Ginibre matrix of size $t$. We will mention this interpretation in Remark 5.4.
>
> > "What would strengthen the work considerably would be experiments using ReLU activation functions."
>
> We thank the reviewer for this relevant suggestion. We have conducted these experiments and the resulting plots show what the Jacobian spectrum looks like during the training of an MLP, similarly as in figure 1 of the paper. We see similar curves with additional noise due to the change of modes. We will add these results.

---

> > ### Author Rebuttal · Reviewer_g5Zs · 2026-04-03
> >
> > I thank the authors for carefully addressing all my points, which mostly were remarks regarding the presentation and partly the interpretation. I keep my high score of this interesting work and hope we will see it published.

---

> > > ### Author Response · Authors · 2026-04-06
> > >
> > > We thank the reviewer for the careful reading of the positive feedback.  We are pleased that our clarifications addressed your concerns.

---

### Official Review · Reviewer_iEHC · 2026-03-11

**Soundness:** 4
**Presentation:** 4
**Significance:** 3
**Originality:** 4
**Overall Recommendation:** 5
**Confidence:** 5

**Summary:**

The paper analyzes implicit bias in gradient-based training via deep Jacobian structure. It identifies two signatures: exponential scaling of ordered singular values with depth and strong spectral separation. Under a fixed-gates view, it derives Lyapunov exponents and shows singular-vector alignment, enabling decoupled singular-value dynamics and explaining emergent low-rank Jacobian structure.

**Compliance With Llm Reviewing Policy:**

Affirmed.

**Final Justification:**

Solid paper. i vote for acceptance.

**Key Questions For Authors:**

Can the techniques in this paper be extended to the ResNet setting, where the Jacobian takes the form  $J= \prod_l (I+D_lW_l)$, as studied in some previous works [4] [5]?

[4] https://proceedings.mlr.press/v89/tarnowski19a.html

[5] https://arxiv.org/abs/1807.11694v1

**Limitations:**

Yea

**Strengths And Weaknesses:**

Pros

1 This is a technically sound paper. The theoretical results are insightful, and the corresponding proofs appear convincing.

2 The assumption that $D$ has {0,1} entries is close to practical settings, for example in ReLU neural networks. Moreover, the results seem likely to extend to other architectures with ReLU-like activations.

3 Importantly, the empirical validation of the theoretical results is impressive.

Cons

The authors assume that $D_l$ and $W_l$ of are independent, but it is not true for the Jacobian matricies of NNs. However, this is not strictly true for the Jacobian matrices of neural networks. That said, I do not consider this to be a major issue, as independence (or asymptotic freeness) assumptions are common in the literature on dynamical isometry of neural network Jacobians, e.g.,  in the series of works by Jeffrey Pennington.

Moreover, I would like to note that the pioneer Leonid Pastur in RMT has several seminal works [1-3] that rigorously study asymptotic freeness for random matrices arising in deep neural networks. These works may provide useful theoretical context and could be discussed in the paper.

[1] Pastur, L. (2022). Eigenvalue distribution of large random matrices arising in deep neural networks: Orthogonal case. Journal of Mathematical Physics, 63(6), 063505.

[2] Pastur, L., & Slavin, V. (2023). On random matrices arising in deep neural networks: General iid case. Random Matrices: Theory and Applications, 12(01), 2250046.

[3] Pastur, L. (2023). The Law of Multiplication of Large Random Matrices Revisited. Journal of Mathematical Physics, Analysis, Geometry (18129471), 19(1).

---

> ### Author Rebuttal · Authors · 2026-03-30
>
> We thank the reviewer for the careful reading and constructive feedback.
>
> > "Can the techniques in this paper be extended to the ResNet setting, where the Jacobian takes the form $J = \prod_\ell (I + W_\ell D_\ell)$, as studied in [4] and [5]?"
>
> Extending our results to the ResNet setting seems non-trivial and would likely require fundamentally different arguments.
> It is important to first distinguish two related but distinct spectral phenomena. Our paper is concerned with exponential depth-scaling of ordered singular values, which is what produces strong spectral separation and ultimately drives the implicit bias mechanism we study.  This is different from the vanishing gradient problem, which concerns the collapse of the overall Jacobian norm. The skip connections in ResNets primarily address the latter: as shown in [4] and [5], with appropriate weight initialization (variance scaling as $1/(NL)$), the singular values of the ResNet Jacobian concentrate around 1 at initialization — achieving dynamical isometry. This concentration is the opposite of the exponential spreading our paper relies on: in the ResNet setting there is no depth-induced separation at initialization, so it seems that Theorem 5.3, Corollary 5.6, and Theorem 6.1 would not carry over.
> From a mathematical standpoint, our proofs rely on the Haar invariance property of Ginibre matrices, which allows us to treat singular vector bases as Haar-distributed. The introduction of the identity term in $(I + W_\ell D_\ell)$ breaks this symmetry, which prevents a direct application of our arguments. Developing the product into multiple sums would only make the expectation tractable, but certainly not the almost sure convergence of the singular vectors, as we would be computing the SVD of a large sum with heavily correlated terms.
>
> That said, [4] and [5] establish that ResNet Jacobians have a well-characterized limiting spectrum, which could serve as a starting point for a different analysis, in which we do not expect depth scaling to be involved. We do not exclude the possibility that other related ideas could be adapted to that setting. We consider this an important and non-trivial direction for future work.
>
> > "There are several seminal works by Pastur [1-3] that rigorously study asymptotic freeness for random matrices arising in deep neural networks. These may provide useful theoretical context."
>
>  We thank the reviewer for pointing out these relevant works, which we will add to the related work section.

---

> > ### Author Rebuttal · Reviewer_iEHC · 2026-04-01
> >
> > I do not require the authors to cite [4] or [5].I asked these questions out of personal interest. This is not a weakness of the paper. Generally speaking, I like this paper and vote for acceptance.

---

> > > ### Author Response · Authors · 2026-04-06
> > >
> > > We thank the reviewer for the careful reading of the positive feedback.  We are pleased that our clarifications addressed your concerns.

---

### Decision · Program_Chairs · 2026-04-30

**Decision:**

Accept (spotlight)

**Comment:**

# Summary

This paper studies the input-output Jacobian spectra of fully-connected networks with fixed gate patterns from the viewpoint of random matrix theory. Specifically, the paper mainly studies a deep model named Fixed-Gate Linear Network (FGLN) whose input-output Jacobian takes the form $J = W_L D_{L-1} W_{L-1} \cdots D_1 W_1$, where the $D_l$ are fixed diagonal matrices and the $W_l$ are network parameters. Under the special case where the diagonal entries of $D_l$ are Bernoulli samples from $\{0, 1\}$, the authors establish depth-induced exponential scaling and separation of singular values, which in turn induces singular-vector alignment in matrix products. From these results, the paper identifies two key assumptions under which the singular-value dynamics of FGLN are decoupled, which mirrors the results for linear neural networks under balanced initialization.

# Comments

All reviewers were very positive about the paper, and the concerns were well addressed by the authors. The reviewers have agreed that this is a strong contribution to the community, expanding our understanding of the evolution of neural network Jacobian spectra. Although the model itself is still a bit stylized (i.e., activation patterns independent of input and parameters), I believe that the technical contributions extend well beyond plain linear neural networks. I recommend acceptance.

Some suggestions that I have, after reading the paper and the discussion myself:
* In line 234, the paper claims that the approximation is independent of $r$, but the summation itself actually depends on $r$. Did the authors mean something along the lines of “much weaker dependence on $r$”?
* In Theorem 6.1, it would be helpful to discuss how restrictive it is to assume that the right singular vector matrix of $A_L$ is the identity matrix (is it without loss of generality?). Also, the statement uses $r$ before assuming its existence, which should be corrected.
* From my understanding, Proposition 7.1 appears to hold for general FGLNs, not just masked linear networks considered in Sections 5 and 6. It would be helpful to put a reminder at the beginning of Section 7.
* I think it would be better to state Remark 7.2 as a separate proposition, and also to emphasize that the result applies to general networks such as multi-mode FGLNs. In its current form, multi-mode FGLNs are defined earlier but are not explicitly used anywhere else in the main text.
* Adding the ReLU network experiments that the authors mentioned in the rebuttal would enhance the paper.